# Ranking environmental degradation trends of plastic marine debris based on physical properties and molecular structure

Kyungjun Min [1], Joseph D. Cuiffi[2] & Robert T. Mathers [1]*

As plastic marine debris continues to accumulate in the oceans, many important questions surround this global dilemma. In particular, how many descriptors would be necessary to model the degradation behavior of ocean plastics or understand if degradation is possible? Here, we report a data-driven approach to elucidate degradation trends of plastic debris by linking abiotic and biotic degradation behavior in seawater with physical properties and molecular structures. The results reveal a hierarchy of predictors to quantify surface erosion as well as combinations of features, like glass transition temperature and hydrophobicity, to classify ocean plastics into fast, medium, and slow degradation categories. Furthermore, to account for weathering and environmental factors, two equations model the influence of seawater temperature and mechanical forces.

[1] Department of Chemistry, Pennsylvania State University, New Kensington, PA 15068, USA. [2] Department of Electro-Mechanical Engineering Technology, Pennsylvania State University, New Kensington, PA 15068, USA. *email: rtm11@psu.edu

During the past several decades, accumulation of plastic in the oceans has emerged as a global challenge[1]. Currently, the severity of plastic marine debris continues to increase and has reached worldwide proportions[2,3]. In spite of the enormous scale, a wide variety of environmental, marine biology, oceanography, ecology, and toxicology investigations have already gleaned important insight on ocean plastics. This includes the arduous task of collecting, sorting, and identifying ocean plastics along with exploring mechanisms for degradation, and examining implications for aquatic life[4–6].

Due to a wide variety of environmental factors, such as exposure to UV radiation, wind, waves, seawater, and bacteria, plastic waste experiences concurrent influences leading to cracking, surface erosion, abrasion, and breakdown to mesoplastic (~5–20 mm), large microplastic (~1–5 mm), small microplastic (~20–999 μm), and nanoplastic (<1 μm) sized pieces[7–9]. This multi-faceted situation has led to questions about size distribution and composition along with inquiries into how bulk properties of plastics change in the ocean[9–11]. Consequently, a large amount of data on physical (i.e., density, surface roughness, weight loss over time), thermal [i.e., melting temperature ($T_m$), glass transition temperature ($T_g$)], and mechanical (i.e., modulus) properties have been measured along with molecular weight changes. Spectroscopic methods, like ATR-FTIR spectroscopy, and chromatographic techniques, such a gel-permeation chromatography (GPC), have greatly assisted in characterization of plastics[12].

On a molecular level, three important degradation mechanisms impact physical and thermal properties of plastics in the ocean[10]. First, depending on surface energy, bacteria colonize surfaces in the ocean and have a propensity for biofilm formation[13–15]. This provides an opportunity for biodegradation in the form of mass loss via surface erosion. While the number of bacteria and microbial enzymes that facilitate surface erosion is enormous[16], a few examples include biodegradation of polyesters (polycaprolactone, PCL)[17], polyamides (Nylon 6)[18], and polyolefins (i.e., polyethylene, PE)[19]. Typical rates for these processes decrease as follows: polyesters > polyamides > polyolefins.

Second, abiotic hydrolysis of functional groups, like esters, carbonates, and amides, severs the large macromolecules that comprise a piece of plastic and thereby reduce molecular weight[4]. This process is facilitated by the alkalinity of seawater (pH range ~8–8.3) and presence of hydroxide (i.e., OH⁻) ions[20]. Based on degradation studies at different temperatures and subsequent calculation of activation energy for abiotic hydrolysis, propensity for degradation depends on functional groups and polymer structure such that PCL (81 kJ mol⁻¹) > bisphenol A polycarbonate (PC) (92 kJ mol⁻¹) > PET (125 kJ mol⁻¹)[21].

Generally, these abiotic and biotic processes proceed slowly and depend on a number of factors, like type of functional group, molecular weight, and surface to volume ratio. Functional groups, like esters, amides, carbonates, and urethanes, allow much faster surface erosion via enzymatic hydrolysis and abiotic hydrolysis than plastics without functional groups, such as PE and polystyrene (PS). Although Nylon degradation is slower than polyesters, biodegradation of nylon rope submerged in the ocean was 1% per month over a 12-month period[22].

Third, exposure to UV radiation and oxygen causes photodegradation[23,24]. These photodegradation processes occurs to a depth of 50–100 μm and result in molecular weight reduction and cracking that facilitates microplastic formation[7]. In addition, as C–H bonds oxidize, the resulting carbonyl groups, like aldehydes and ketones, facilitate a "higher coverage of biofilms"[25]. Since these photodegradation processes involve a radical mechanism, the likelihood of photo-initiated C–H oxidation and

chain scission depend on polymer structure. As a result, most commercial plastics contain additives such as antioxidants and light stabilizers that delay degradation[26,27]. In general terms, polymers without tertiary hydrogens, like poly(methyl methacrylate) (PMMA) and polytetrafluoroethylene (PTFE), are often highly stable. In contrast, others are moderately stable (PET, PC) or poorly stable due to the presence of tertiary C–H bonds [PS, poly(vinyl chloride) (PVC)], allylic C–H bonds [polyisoprene (PI), polybutadiene (PBD)], C=O bonds [polyamides (PA), polyurethanes (PU)], and catalyst residues (PE, polypropylene (PP)][26]. Overall, a comparison of plastics with and without tertiary C–H bonds reveals reactivity (i.e., bond dissociation energies) decreases as follows: PVC > PS > PP > PE[4].

Although the enormous scale of plastic debris in the ocean is daunting, many informative publications have laid a foundation for understanding the scope of this dynamic, global problem. As a result, opportunities have emerged to illuminate important features that influence the most common types of degradation mechanisms. Consequently, we hypothesize a combination of experimental data and computational predictions could translate polymer structure on a molecular level into a predictive model that addresses unanswered questions regarding the viability of degradation in the ocean. Here, thorough analysis of polymer structure, composition, physical properties, and degradation data, we predict a hierarchy of features that regulate degradation. As such, the following investigation starts with a simple comparison (Fig. 1) and systematically increases complexity (Figs. 2–3) followed by further refinement with machine learning (ML) (Figs. 4–5). Then, inspired by the importance of hydrophobicity and $T_g$, two equations for quantifying surface erosion are presented.

## Results

**Database.** Initially, constructing a database delineated polymer structure, physical properties, and experimental degradation data in the literature. As a caveat, the applicability of biodegradation tests span a wide range, and biodegradation in one environment (i.e., soil) may not always transfer to other scenarios (i.e., oceans)[28]. As a result, preference has been given to ocean studies, those using seawater from the ocean in a laboratory, artificial seawater with marine bacteria, or enzymes.

Over 110 polymers including polyesters with linear[29], branched[30], and cyclic[31] structures as well as polyacetals[32], PA[33], polyacrylamides, PC[34], polyethers[35], PE[10,25], PP[22,36], polysiloxanes[15], PS[37], PU[15,38], and PVC[39] were investigated (Supplementary Fig. 1). Plastics in the database included commercial samples (69) and those made in a laboratory (46). Polymers were categorized by class (i.e., type of polymer), specimen (i.e., films, powders), physical attributes (i.e., mass, volume, surface to volume ratio), and experimental parameters (i.e., time in seawater, temperature). Weight loss during exposure to seawater as well as abiotic or biotic conditions was also recorded. Additionally, molecular level descriptors and bulk polymer descriptors differentiated each polymer.

Bulk property descriptors included density, weight-average molecular weight ($M_w$), number-average molecular weight ($M_n$), dispersity ($M_w/M_n$), $T_g$, melting temperature ($T_m$), % crystallinity, and enthalpy of melting (i.e., amount of energy required in J g⁻¹). Molecular level descriptors included types of carbon, oxygen, and nitrogen atoms using the concept of hybridization (i.e., sp³, sp²) and the % of these atoms in the polymer. To capture architectural features on the molecular level, the database denoted the number of hydrogens per monomer, number of $CH_3$, $CH_2$, and CH groups per monomer, the number of cyclic rings, and % atoms in cyclic rings. To quantify the oil-like or water-repellent attribute of

each polymer on a continuum, a concept termed hydrophobicity was investigated. Overall, the database contained >110 polymer samples with >5000 descriptors.

### Hydrophobicity

In Fig. 1a, quantifying hydrophobicity involved a molecular level method that combines theory, simulation, and experimental validation[40-42]. The theory was inspired by pharmaceutical advances in determining the solubility of drug-like molecules with computational octanol-water partition coefficients (Log$P$)[43]. Based on the Log$P$ equation in Fig. 1a, both negative and positive values are possible. Negative Log$P$ values predict water solubility, polymers that swell in water, or polymers that demonstrate a propensity to absorb water while positive values predict insolubility in water. Using molecular dynamics (MD) simulation to minimize energy of molecular models followed by calculation of surface area (SA) allows comparison of different polymers.

In addition to the thermodynamic significance of octanol-water partition coefficients (Eq. 1), which describes the free energy ($\Delta G_{transfer}$) required to transfer a molecule from water to octanol[44], this strategy underscores the important role of SA rather than volume[45]. Consequently, Log$P$(SA)$^{-1}$ values have provided a molecular level strategy to predict solubility and structure for applications involving crystallization driven self-assembly (CDSA) and polymerization induced self-assembly

(PISA)[46-48].

$$LogP = -\frac{\Delta G_{transfer}}{RT \ln 10} \quad (1)$$

### Screening features

After creation of the database, a question arose regarding which molecular and bulk descriptors would provide the best prediction of degradation in the ocean. Accordingly, all the features from the database were screened for trends using data-analytics approaches, such as correlation matrices (Supplementary Fig. 2). Out of this initial pre-screening, seven attributes seemed promising: density, molecular weight, $T_g$, % crystallinity, enthalpy of melting, % sp$^3$ carbons, and Log$P$(SA)$^{-1}$. Interestingly, Log$P$(SA)$^{-1}$ values have sensitivity to hybridization (i.e., % sp$^3$ and sp$^2$ carbons), density, large numbers of atoms (i.e., H, C, N, O, Si, P, S, Cl, Br, F) and how these atoms are connected. Consequently, this descriptor applied to more polymers than single features, like % nitrogen atoms, which works well for PA, or % sp$^3$ carbons, which was informative but better suited for a single class of polymers, like polyesters. As a result of the correlation between Log$P$(SA)$^{-1}$ and other features (Supplementary Fig. 3), the list of seven possible predictors shortened to five: molecular weight, $T_g$, % crystallinity, enthalpy of melting, and Log$P$(SA)$^{-1}$.

After pre-screening the database, we wondered how many features would be necessary to understand the nearly overwhelming complexity of plastic degradation in the ocean. As a

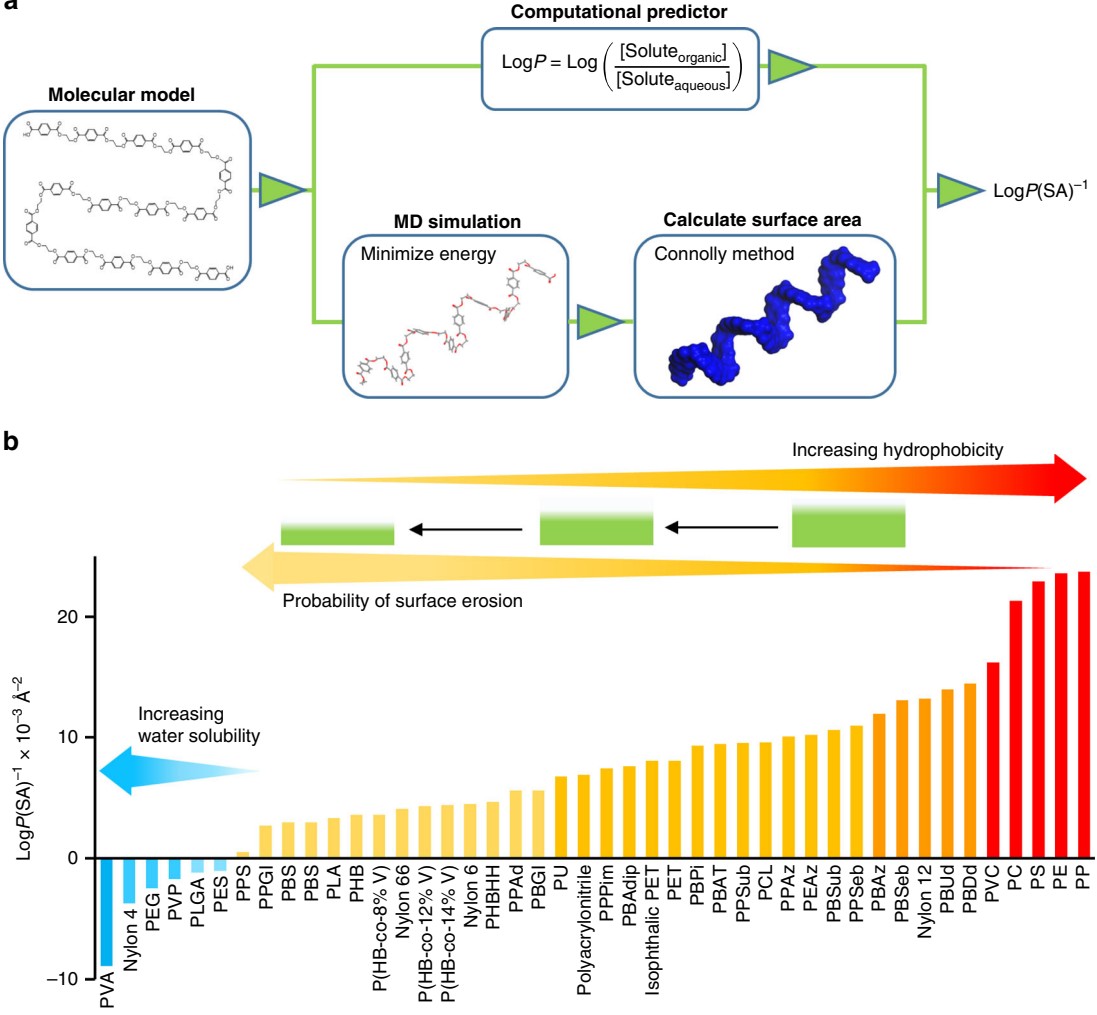

**Fig. 1 Plastics cover a wide range of hydrophobicity. a** Flow chart for calculating hydrophobicity, **b** range of Log$P$(SA)$^{-1}$ values for various plastics.

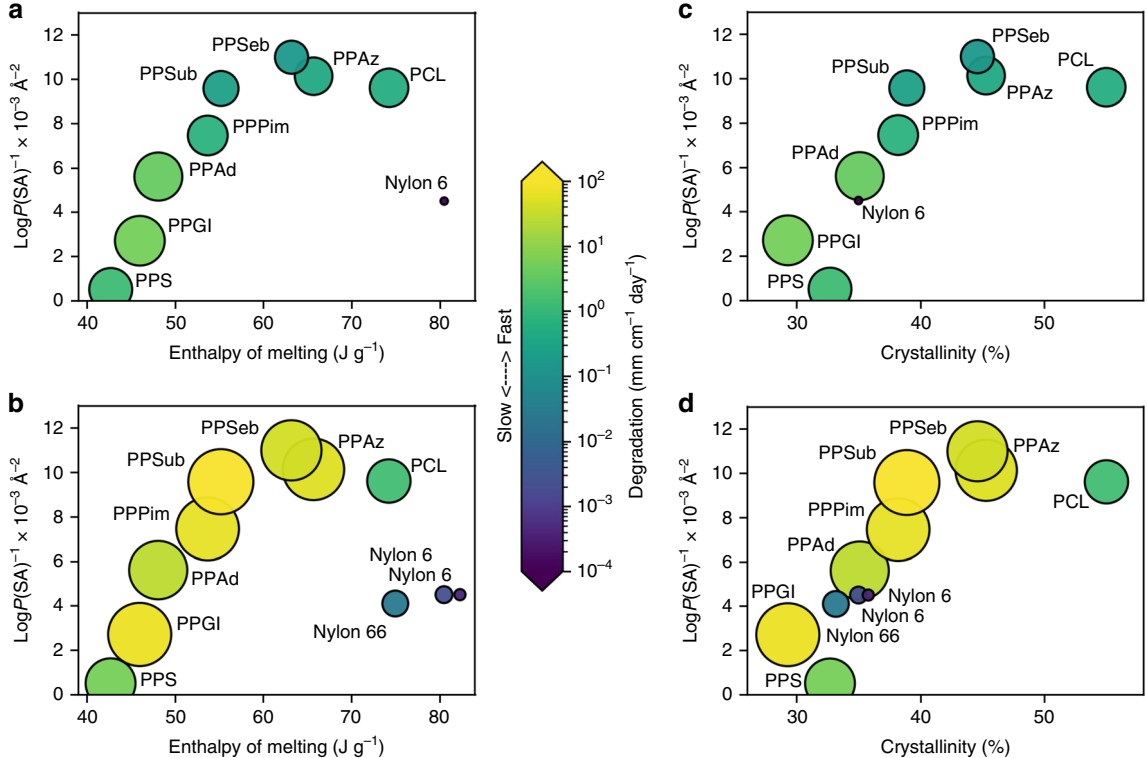

**Fig. 2 Influence of crystallinity and hydrophobicity on degradation.** Computational $LogP(SA)^{-1}$ values versus **a** enthalpy of melting for abiotic hydrolysis, **b** enthalpy of melting for biotic processes, **c** % crystallinity for abiotic hydrolysis, **d** % crystallinity for biotic processes. Size of circles and color corresponds to surface erosion in mg cm$^{-2}$ day$^{-1}$ in artificial seawater.

result, the complexity of analysis systematically increased in Figs. 1–5. For instance, Fig. 1 explores a straightforward evaluation of molecular structure with one feature. Then, Fig. 2 compares degradation data under controlled condition in a laboratory setting with two features. Figure 3 widens the number of samples by comparing laboratory and ocean conditions on a 5-tier scale. Then, Figs. 4 and 5 use a ML method to further explore and refine this question.

Figure 1b arranges common types of plastics found in the ocean and a wide variety of other examples according to $LogP(SA)^{-1}$ values. These initial efforts to investigate molecular structure indicate functional groups substantially lower the hydrophobicity relative to polyolefins. For example, Nylon 6 [$LogP(SA)^{-1} = 0.0045$ Å$^{-2}$] and PCL [$LogP(SA)^{-1} = 0.0096$ Å$^{-2}$] were considerably less than PE [$LogP(SA)^{-1} = 0.0236$ Å$^{-2}$]. Furthermore, this convenient method helped sort plastics into several groups.

The first group consists of water-soluble plastics [$LogP(SA)^{-1} < 0$ Å$^{-2}$] in Fig. 1b. These types, like poly(ethylene glycol) (PEG) or poly(vinyl alcohol) (PVA), have polar functional groups (i.e., OH groups) that degrade via microbial oxidation[49]. Alternatively, other functional groups such as amides in Nylon 4 degrade through biotic hydrolysis[50]. A second group in Fig. 1b comprises insoluble plastics [$0 < LogP(SA)^{-1} < \sim 0.013$ Å$^{-2}$] susceptible to surface erosion via biodegradation, abiotic hydrolysis through exposure to seawater, and photodegradation. Within this category, the propensity for polyester surface erosion correlates with hydrophobicity when the $T_g$ values < ocean temperature. A similar trend was noted for nylons as proclivity to degrade decreased accordingly: Nylon 4 > Nylon 6 > Nylon 12[18,50]. The third group [$LogP(SA)^{-1} > \sim 0.015$ Å$^{-2}$] in Fig. 1b corresponds to the most hydrophobic plastics that may not have functional groups for abiotic hydrolysis but most likely have a large percentage of C–H bonds susceptible to photodegradation. In

addition to oxidation via photo-initiated processes, extremely slow surface erosion is observed for PE and PP. Recent studies confirm that plastics produced in the highest volume, like PE and PP, make up a disproportionate percentage of ocean plastics near the sea surface[5]. Interestingly, $LogP(SA)^{-1}$ values for these very hydrophobic plastics correspond to lower densities (Supplementary Fig. 3) that would enable floating near the sea surface.

While the ranking in Fig. 1 generally correlates with proclivity for polyester degradation, plastics with $T_g$ values > ocean temperature, like PLA, PLLA, and PET, degrade more slowly than expected[30]. For instance, although PLA degrades under composting conditions, in seawater degradation proceeds very slowly[28]. This highlights the need for multiple metrics to understand degradation in the ocean. As a result, crystallinity, enthalpy of melting, $T_g$, molecular weight, and $LogP(SA)^{-1}$ values were investigated in pairs to find patterns of degradation.

**Crystallinity**. To further explore functional groups and hydrophobicity trends in Figs. 1, 2 compares crystallinity and enthalpy of melting with $LogP(SA)^{-1}$ values for abiotic and biotic conditions. As denoted by the size of the circles in Fig. 2, surface erosion was calculated using surface area of each plastic object (SA$_{bulk}$), mass loss, and number of days in seawater. To achieve a systematic variety of hydrophobicity values, the number of hydrophobic methylene (CH$_2$) units in the monomer structures ranged from 5 for poly(propylene succinate) (PPS) to 11 for poly(propylene sebacate) (PPSeb).

In Fig. 2, several meaningful observations are worth mentioning. First, enzymatic degradation of polyesters with $T_g$ values below ocean temperature was faster than abiotic hydrolysis. Although Nylon 6 exhibited a similar trend[18], further comparison with other plastics was difficult due to the lack of studies

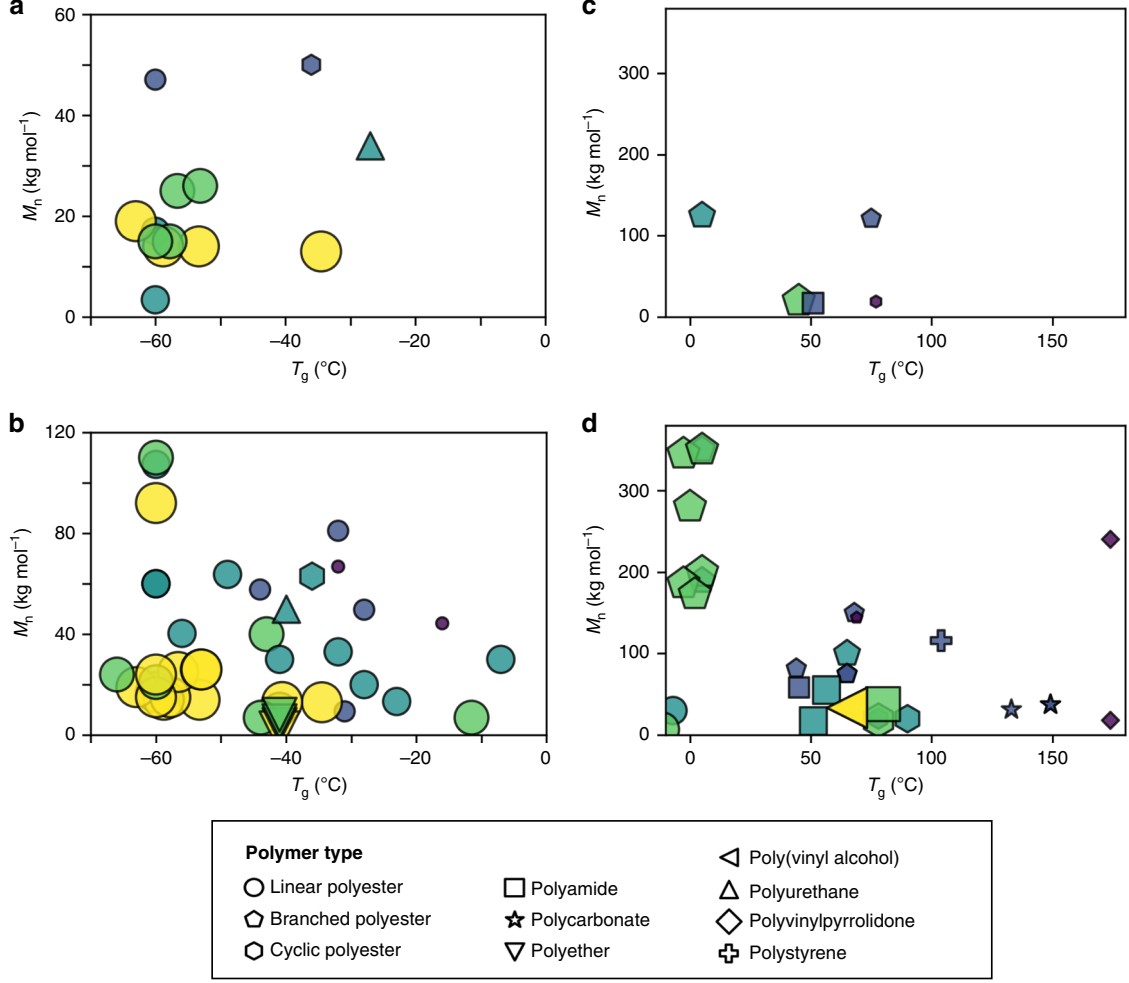

**Fig. 3 Effect of $T_g$ and $M_n$ on polymer degradation. a** Abiotic processes for $T_g$ values < 0 °C, **b** biotic processes and photo-initiated oxidation for $T_g$ values < 0 °C, **c** abiotic processes for $T_g$ values > 0 °C, **d** biotic processes and photo-initiated oxidation for $T_g$ values > 0 °C. Degradation categories depicted on 5-tier scale and shown by size of data point and color (yellow > light green > green > blue > purple).

comparing abiotic and biotic degradation under similar conditions. Interestingly, while laboratory experiments for polyesters in Fig. 2 fail to account for weathering processes and mechanical forces in the ocean, controlled conditions help separate the influence of abiotic hydrolysis from biodegradation and photo-initiated C–H bond oxidation. If abiotic hydrolysis, biodegradation, and photo-initiated processes occur simultaneously, then decreases in molecular weight via abiotic hydrolysis or photo-initiated reactions could facilitate biotic processes while enzymatic hydrolysis might promote abiotic hydrolysis. Second, abiotic hydrolysis in Fig. 2a and c appears more sensitive to increases in hydrophobicity, enthalpy of melting, and % crystallinity than biotic processes. For instance, the largest abiotic hydrolysis rates for poly(propylene glutarate) (PPGl) and poly(propylene adipate) (PPAd) slowed as hydrophobicity (Log$P$(SA)$^{-1}$ > 0.007 A$^{-2}$) and enthalpy values (>50 J g$^{-1}$) increased. In contrast, biotic processes demonstrate faster rates for more hydrophobic polyesters, like poly(propylene pimelate) (PPPim) and poly(propylene suberate) (PPSub). Third, comparison of polyesters and PA (i.e., Nylon 6, Nylon 6,6) indicate biotic and abiotic processes still occur for semicrystalline plastics but crystallinity will slow these processes. A comparison of PLA and PLLA (Supplementary Fig. 4) indicate the increased % crystallinity of PLLA slows surface erosion. Although % crystallinity, enthalpy of melting and $T_m$ values are all informative, crystallinity and enthalpy of melting allow an easier comparison of polyesters and PA (i.e., Nylon 6, Nylon 6,6) than $T_m$ values. For example, the relationship between $T_m$ values and degradation show opposite trends for polyesters and PA. As such, degradation decreases as follows: Nylon 4 ($T_m$ ~ 267 °C) > Nylon 6,6 ($T_m$ ~ 264 °C) > Nylon 6 ($T_m$ ~ 220 °C)[18,50]. In contrast, polyesters with lower $T_m$ values, such as PCL ($T_m$ ~ 60 °C), show faster degradation than poly(ethylene succinate) (PES) ($T_m$ ~ 104 °C).

To account for surface to volume ratio and time under controlled conditions, Fig. 2 compares degradation data for polymer films using units of mg cm$^{-2}$ day$^{-1}$. In many cases, mg cm$^{-2}$ day$^{-1}$ values were not reported but could be calculated when dimension and weight of samples were given with the experimental section. However, the wide variety of experimental parameters (i.e. temperature, films, powders, discs), environmental conditions in the ocean, as well various methods for reporting weight loss [i.e., %, mg cm$^{-2}$ day$^{-1}$, and BOD (% day$^{-1}$)] makes comparison of data difficult. As a result, in Figs. 3–5, a second strategy was devised to compare experiments under controlled conditions in a laboratory with ocean studies. This method converted various weight loss values into 3-tier categories (slow, medium, fast) and 5-tier categories (very slow, slow, medium, fast, very fast). As a reference, poly(butylene adipate) (PBAdip), which appeared in several studies was assigned a medium value.

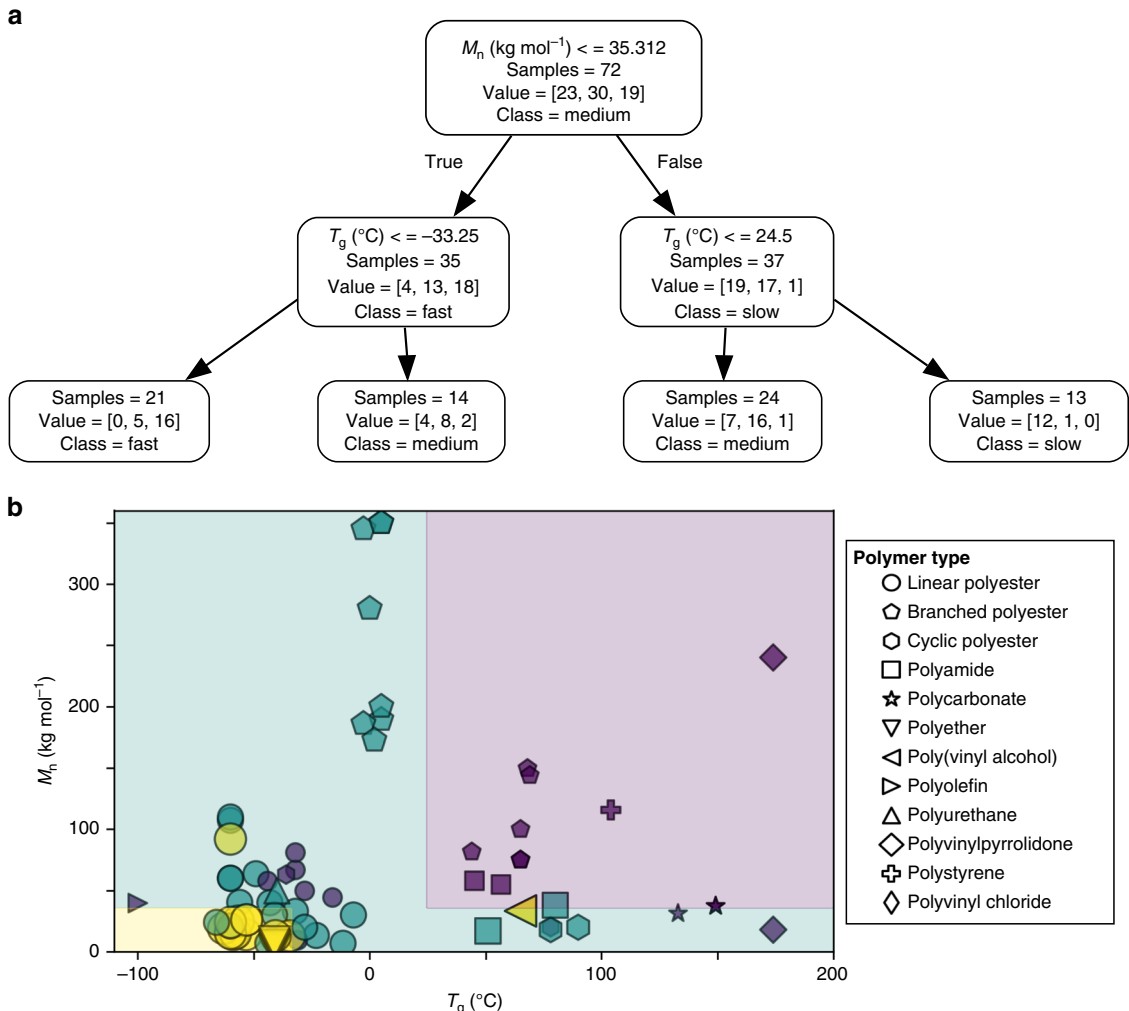

**Fig. 4 Machine learning analysis with 2-levels of classification. a** Decision tree based on molecular weight and $T_g$. Within each box of the tree, value = [x, y, z] corresponds to [slow, medium, fast]. **b** Graphical representation of decision tree. The shaded areas reflect prediction zones that correspond to fast (yellow), medium (green), and slow (purple) degradation.

**$T_g$ and molecular weight.** Figure 3 investigated the combined effect of $T_g$ and molecular weight on degradation. In the context of photo-initiated C–H bond oxidation, abiotic hydrolysis, and biotic activity, several observations emerged. First, degradation trends paralleled $T_g$ values and decreased accordingly: linear polymers (i.e., PCL) > branched polymers with methyl groups (i.e., PHB and PHBV) > polymers with cyclic rings and functional groups in polymer chain (i.e., PBAT, PET, PC) > polymers with cyclic rings and all carbon atoms in polymer chain [i.e., PS, poly (vinyl pyrrolidone)]. These trends suggest degradation occurs more rapidly with $T_g$ values below ocean temperature. However, some plastics without functional groups, such as polyolefins, exhibit very slow degradation even though $T_g$ values are quite low. Furthermore, additives in commercial polyolefins slow degradation for PE (0.45 wt. % month$^{-1}$) and PP (0.39 wt. % month$^{-1}$)[22].

Second, the fastest abiotic hydrolysis occurred for molecular weights below ~25 kg mol$^{-1}$. However, when $T_g$ < ocean temperature, enzymatic activity degraded PHB ($T_g$ ~ 2–5 °C) reasonably well even when molecular weight was 200–700 kg mol$^{-1}$. Third, Fig. 3 provides a framework for estimating plastics based on two common experimental measurements, namely molecular weight and $T_g$. However, this framework works best for comparing polymers with either all positive or all negative

Log$P$(SA)$^{-1}$ values. In Fig. 3d, the negative Log$P$(SA)$^{-1}$ value for the polyol (i.e. polyvinyl alcohol), which is shown by a left-facing yellow triangle, seems out of place when superimposed on plastics with positive Log$P$(SA)$^{-1}$ values. This illustrates the difficulty in comparing negative and positive Log$P$(SA)$^{-1}$ values on a graph of molecular weight versus $T_g$. Additionally, another example of this challenge occurred in Fig. 3c for $T_g$ < ocean temperature when comparing the negative Log$P$(SA)$^{-1}$ value for polyethers (i.e. PEG) with the positive Log$P$(SA)$^{-1}$ values for linear polyesters.

**Machine learning.** In order to further investigate biodegradation trends, ML analysis of physical property data was conducted. Although the current data set needs more PC, PA, and PU samples to develop high-accuracy validated prediction models, decision trees were explored because of their value in visualizing information gained from categorizing data. Figures 4 and 5 show decision trees that classify polymers using the following features: $M_n$, $T_g$, enthalpy of melting, and Log$P$(SA)$^{-1}$.

The decision trees were trained on the data using Gini impurity and manual limiting of the depth to 2–3 levels to avoid overfitting. Accuracy of the decision tree model on the training data (Supplementary Fig. 5) increased from 72.2% with two-levels

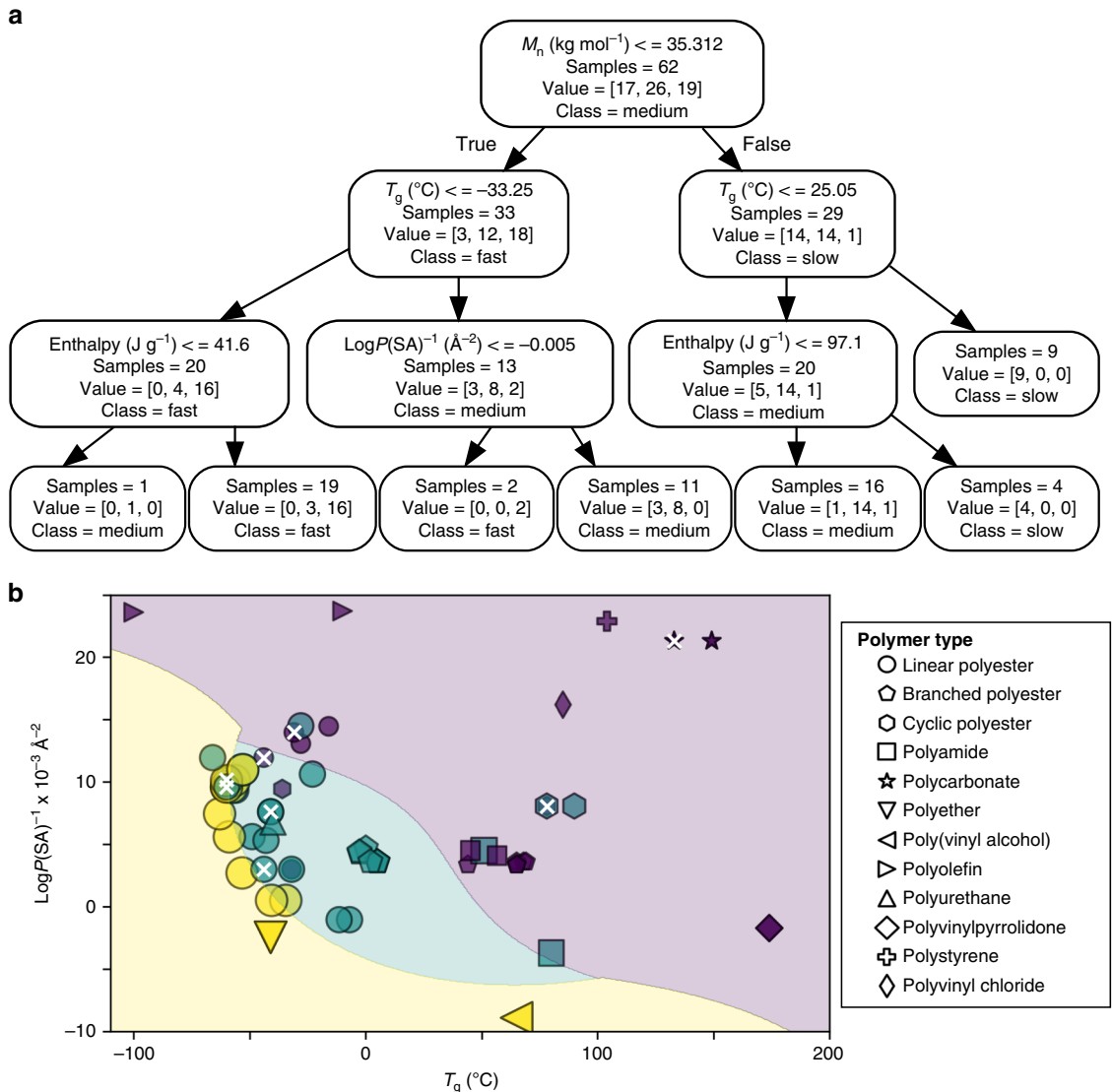

**Fig. 5 Machine learning results with 3-levels of classification. a** Decision tree based on molecular weight, enthalpy, $\text{Log}P(\text{SA})^{-1}$, and $T_g$. Within each box of the tree, value = [x, y, z] corresponds to [slow, medium, fast]. **b** Graphical representation of two of the four features. The shaded areas, added to visually generalize the degradation regions, correspond to fast (yellow), medium (green), and slow (purple) degradation. Incorrect predictions from the classification tree are denoted with an 'x'.

containing two features to 87.1% with three-levels comprising four features. Due to the relatively limited number of samples, applying ten-fold cross-validation to the models provided an accuracy of the 57.8% for the two-level model and 63.2% for the three-level model. In either case, the models avoided incorrectly classifying a fast degradation process as slow degradation and vice versa. Based on these results, two to four features are powerful predictors for degradation categories (i.e., fast, medium, and slow) for a wide variety of polymers. Even with two features, like molecular weight versus $T_g$ or $\text{Log}P(\text{SA})^{-1}$ versus $T_g$, the compelling results underscore the connection between environmental degradation and structure-property relationships. Interestingly, the division between slow and medium degradation in Fig. 4 approximates ocean temperature.

Figure 5a shows additional depth to the decision tree with three-levels of classification. This tree uses four predictor features and provides improved accuracy over the two-level tree. Given the wide variety of experimental parameters in the database, some incorrect predictions are expected. In the case of Fig. 4, most

errors resulted for plastics, like polyolefins, when $\text{Log}P(\text{SA})^{-1}$ exceeded ~0.010 $\text{Å}^{-2}$ or when $\text{Log}P(\text{SA})^{-1} < 0$ as demonstrated by water-soluble polymers, like PVA. However, moving from the two-level tree in Fig. 4 to the three-level tree, the number of inaccurate predictions decreased from 20 to 8, respectively. This suggests Fig. 4 applies to a narrower window of $\text{Log}P(\text{SA})^{-1}$ values than Fig. 5.

In order to visualize these errors, Fig. 5b shows the eight incorrect predictions of the training set on plot of $\text{Log}P(\text{SA})^{-1}$ versus $T_g$. Although this plot reasonably splits into zones of fast, medium, and slow degradation, most errors congregate on the fast-medium or medium-slow border where degradation categories merge together. As a result, these inevitable boundary errors differ from conflicting literature data. To elaborate, the data contains instances were differences in environmental conditions as well as comparison of commercial materials with those produced in a laboratory produced a range of degradation behavior. As denoted by a symbol containing an 'x' in the slow category, this was especially true for PET and PC.

To illustrate the challenge of assessing diverse environmental conditions, variances in temperature, ocean conditions, and laboratory studies resulted in ranking of PET degradation from very slow, slow, medium, and fast on a five-tier scale and slow to medium on a three-tier scale. The dilemma of deciding which data trend is the most appropriate highlights the need for a data-driven method to analysis multiple possibilities. During ML, we noticed PET was ranked as medium in the two-level decision tree and slow in a three-level tree. Since the three-level tree in Fig. 5 produced less incorrect predictions than molecular weight and $T_g$ (Supplementary Fig. 6), the location of PET in the slow category of Fig. 5b is more appropriate for commercial plastics than the medium category in Figs. 3c and 4b. Moreover, assessment of PET as slow in Fig. 5b agrees with observations of ~20 year old PET in marine environments[31].

## Discussion

In order to understand the applicability of trends in Figs. 4 and 5, data analysis involved the following considerations: First, to cover a wide range of environmental conditions, data collection included temperatures ranging from ~0 °C[51] to >30 °C[34,52], shallow ocean depths (1–10 m)[22,53], deep seawater (~300 m to >600 m)[17], and simulated deep sea pressure[54]. Second, this study focuses on plastics in direct contact with either real seawater or artificial seawater. As a result, certain scenarios, like microplastics that wash onto beaches due to mechanical action of waves and weather on beaches in the dry state or cycle back and forth between the ocean and beach, exceeds the limit of the current database. Third, regarding the presence of bacteria that could potentially result in biodegradation, coastal regions and open oceans to a depth of 225 m have similar number of cells[55]. Consequently, degradation trends in Figs. 4 and 5 could apply to both coastal and open ocean. Fourth, since $T_g$ values exhibit sensitivity to polymer structure[56], molecular weight (i.e., Flory-Fox equation), crosslinking[57], and plasticizers[58], this metric has some comprehensive potential. Differences in heating rates during $T_g$ measurements as well as small quantities of plasticizer introduces variability in the data, but this error is nominal compared to the breath of the categories in Figs. 4 and 5. Furthermore, due to the availability of data, Figs. 4 and 5 includes commercial samples of virgin PVC[39].

The influence of weathering on plastic debris represents a complex issue that depends on a number of parameters, like sample depth, temperature, mechanical forces, and sunlight[10,25,59]. Our analysis of polyesters and PA indicates certain environmental parameters, like seawater temperature ($T_{water}$) and sample depth, have potential to speed up or slow degradation. For instance, examination of PHBV[53] in a coastal area yielded a relationship between surface erosion and temperature and these quantities decreased with increasing ocean depth (Supplementary Fig. 7). Further comparison of PHBV in a coastal region[53] with PHBV in deep sea conditions[17] as well as PCL[60], PLA[61,62], and Nylon[18,22] indicates increasing $T_{water}$ will increase surface erosion (Supplementary Fig. 8). In addition, the magnitude of this temperature effect, as reflected by the slope, depends on type of plastic (PCL > PHBV in coastal area > PHBV in deep sea > Nylon 6 > PLA) and scales with $(T_{water} − T_g)(LogP)^{-1}(SA)$ (Supplementary Fig. 9).

The relationship between surface erosion rates ($k$) and physical properties in Eq. 2 model data from polyesters and PA with $LogP(SA)^{-1} > 0$ and enthalpy of melting $< 85\,J\,g^{-1}$. Essentially, Eq. 2 depends on $(T_{water} − T_g)(LogP)^{-1}(SA)$ and predicts the slope of surface erosion versus temperature with units of $mg\,cm^{-2}\,day^{-1}$ $°C^{-1}$. We hypothesize predictions extend to other polymers containing functional groups with carbonyls (C=O), such as PC and

PU. To test this hypothesis, predictions for PC, PU, and PET (Supplementary Table 1) seem reasonable compared to PCL, PHBV, Nylon 6, and PLA. As a caveat, the intent of Eq. 2 focuses on amorphous or semi-crystalline polymers (enthalpy of melting $< ~90\,J\,g^{-1}$) with $LogP(SA)^{-1} > 0$. Outside of these parameters, Eq. 2 overestimates $k$ for certain polyesters with larger enthalpy of melting values, such as PBS (~132 J g⁻¹)[60] or PBSeb (~125 J g⁻¹)[29].

$$\text{rate of surface erosion}(k) = \exp\left(\left(\frac{T_{water} − T_g}{\frac{LogP}{SA}}\right) − 28795\right)/4177.3$$

(2)

In a preliminary effort to expand upon Eq. 2 and capture the multi-faceted processes that influence degradation, a simple model in Eq. 3 is proposed. Inspired by efforts to describe weathering[59,63,64], this model assumes the total amount of erosion ($E_{total}$) depends on abiotic processes, biotic processes, seawater temperature ($T_{water}$), and mechanical forces ($E_{waves}$). As such, $k$ from Eq. 2 describes the rate of abiotic and biotic processes and $b$ is the y-intercept in the absence of mechanical forces. To calculate $E_{waves}$, the difference in surface erosion between ocean conditions and sheltered locations is proposed. For example, surface erosion of PHBV increased when exposed to coastal locations[53] ($E_{waves} = 0.017\,mg\,cm^{-2}\,day^{-1}$) and an estuary[65] ($E_{waves} = 0.005\,mg\,cm^{-2}\,day^{-1}$) compared to sheltered mangroves. Although more literature data is needed to further explore the limitations of Eq. 3, initial data analysis (Supplementary Table 2) serves as a starting point for future discussions.

$$E_{total} = kT_{water} + b + E_{waves}$$

(3)

In summary, the challenging complexity of plastic degradation in the ocean has been addressed via a database that summarizes available structure-property information data in combination with degradation data. Analysis of various strategies with increasing levels of sophistication resulted in a systematic structure-property investigation of ocean-based degradation. These approaches started with a simple, convenient overview in Fig. 1 and progressed to laboratory conditions in Fig. 2. Then, Fig. 3 involves a data-analytics approach to evaluate a wider range of experimental conditions (i.e., laboratory, ocean) using five-tier categories. Finally, ML refined the analysis in Fig. 3 and provided two-level (Fig. 4) and three-level (Fig. 5) classification trees as well as boundaries between fast, medium, and slow categories. To elaborate on the quantitative and qualitative aspects of Figs. 3–5, Eq. 2 quantified the rate of surface erosion rate ($k$) as a function of temperature.

The outcome of these strategies offers the following benefits: First, in Fig. 1, functional groups, like carbonates, esters, and amides, lower the magnitude of $LogP(SA)^{-1}$ values relative to PE and facilitate abiotic hydrolysis and biotic pathways for degradation. In contrast, larger $LogP(SA)^{-1}$ values (i.e., $> 0.015\,Å^{-2}$) indicate a substantial fraction of C–H bonds in the polymer structure. Nonetheless, even though photo-initiated C–H bond oxidation is feasible, the presence of additives (i.e., antioxidants, light stabilizers)[27] will delay degradation. Second, quantitative evaluation of abiotic and biotic processes under controlled conditions in Fig. 2 indicates that biotic processes are often much faster. In addition, abiotic hydrolysis of polyesters was more sensitive to increases in $LogP(SA)^{-1}$ and crystallinity than biodegradation. Third, in Fig. 3, degradation of plastics with heteroatoms (i.e., O, N) substantially slowed as $T_g$ values increased above ocean temperature. For example, slower degradation was noted for PLA ($T_g ~ 65\,°C$) and PET ($T_g ~ 78\,°C$) compared to PBAT ($T_g ~ −36\,°C$). Fourth, a comparison of Figs. 3–5 suggests a hierarchy of features for predicting the likelihood of degradation. In this regard, $LogP(SA)^{-1}$, which reflects composition, may

be more useful than molecular weight when comparing water-soluble and water insoluble polymers. Furthermore, $\text{Log}P(\text{SA})^{-1}$, molecular weight, and $T_\text{g}$ apply to both amorphous and partially crystalline (i.e., semicrystalline) plastics whereas Fig. 2 focused on parameters that relate to semicrystalline polymers (i.e., % crystallinity and enthalpy of melting).

Moving forward, we propose data-driven, ML techniques, such as the classification trees in Figs. 4–5, inform predictive models like Eq. 2 by identifying the important physical property parameters. Although a larger variety of PA, PS, and PU would be helpful in Figs. 4–5, the k-fold cross validation method indicates even this moderately sized database is sufficient. One area that needs further investigation is the issue of mechanical forces on rate of degradation.

While efforts at sustainable development of plastics has increased in the last decade[66,67], these efforts struggle to meet the growing challenge of plastic waste[68]. Consequently, we emphasize the need for recycling to reduce the global carbon footprint and highlight strategies aimed to accelerate degradation[68]. These include incorporation of "weak links" in the polymer that undergo abiotic hydrolysis faster than the rest of the plastic[69], blends with water-soluble polymers, and additives that promote photo-initiated oxidation.

## Methods

**Terminology**. To avoid confusion, the term polymer refers to a single molecule composed of many units. Models of these polymers are shown in Supplementary Fig. 1. In contrast, plastic refers to a bulk material composed of numerous polymers. Surface erosion refers to mass loss over time for a given $\text{SA}_\text{bulk}$ in units of mg cm$^{-2}$ day$^{-1}$ [37]. The term feature describes variables used in ML.

**Hydrophobicity**. Calculation of $\text{Log}P(\text{SA})^{-1}$ values was determined with Materials Studio 2019. $\text{Log}P$ values were extracted from the QSAR menu using the ALogP98 option. Since plastics in the database lack ionizable functional groups, like amines or phenols, changes to $\text{Log}P$ values due to seawater was assumed relatively insignificant. Connolly SA was calculated with a 1.40 Å probe after conducting a MD simulation of molecular models. The Forcite Geometry Optimization employed a Smart algorithm and COMPASS II forcefield to minimize the energy below certain specifications. Convergence tolerance for the Smart algorithm included a 1.0e-4 kcal mol$^{-1}$ energy convergence, a 0.005 kcal mol$^{-1}$ Å$^{-1}$ force convergence, and a 5.0e-5 Å displacement convergence. To improve accuracy of $\text{Log}P(\text{SA})^{-1}$ values, multiple models ranging from 10, 12, and 14-monomer units were averaged.

**Calculation of surface erosion**. Calculation of mg cm$^{-2}$ day$^{-1}$ for sample films in the database was apprehended by dividing weight loss by $\text{SA}_\text{bulk}$ and the number of days that the films were exposed to seawater. The mg cm$^{-2}$ day$^{-1}$ values under biotic conditions are assumed to have a small contribution from abiotic hydrolysis. In the case of PLA and PLLA[30,61], mg cm$^{-2}$ day$^{-1}$ values are assumed to include the effect of mass loss from autocatalytic hydrolysis.

**Categories**. Surface erosion data in mg cm$^{-2}$ day$^{-1}$ or biochemical oxygen demand (BOD) values in % day$^{-1}$ were converted to categories in the following manner. For 5-tier categories, very slow represented 0–2% BOD day$^{-1}$ and 0–0.0003 mg cm$^{-2}$ day$^{-1}$, slow corresponded to 2–4% BOD day$^{-1}$ and 0.0003–0.003 mg cm$^{-2}$ day$^{-1}$, medium represented 4–6% BOD day$^{-1}$ and 0.003–0.03 mg cm$^{-2}$ day$^{-1}$, fast denoted 6–8% BOD day$^{-1}$ and 0.03–0.3 mg cm$^{-2}$ day$^{-1}$, and very fast signified degradation processes > 8% BOD day$^{-1}$ and > 0.3 mg cm$^{-2}$ day$^{-1}$. For three-tier categories, slow denoted 0–4 % BOD day$^{-1}$ and 0–0.003 mg cm$^{-2}$ day$^{-1}$, medium represented 4–8% BOD day$^{-1}$ and 0.003–0.3 mg cm$^{-2}$ day$^{-1}$, and fast signified >8% BOD day$^{-1}$ and >0.3 mg cm$^{-2}$ day$^{-1}$. Some mg cm$^{-2}$ day$^{-1}$ values from laboratory experiments at higher temperatures were adjusted to fit ocean conditions. As a reference, poly(butylene adipate) (PBAdip) was ranked as medium for five-tier and three-tier categories.

**Treatment of missing data**. In many cases, characterization data from the literature was incomplete. Missing data was supplemented from databases (i.e., www.polymerdatabase.com) or extrapolated from literature values. For density determination (used to calculate mass for surface erosion), calibration curves (Supplementary Fig. 10) for polyesters, nylons, and PHBV were created.

**Data processing and ML**. Data processing, visualization, and ML was performed using the Anaconda python distribution (www.anaconda.com) software (python v3.7.1) (Supplementary Codes 1 and 2), specifically the packages from SciPy[70]

including Pandas, Matplotlib, and scikit-learn (v0.21.2). For the classification tree learning, the Gini impurity index was used for information gain, and the maximum depth was set manually at two and three levels. For the shaded regions in Fig. 5b, a support vector machine algorithm was used to classify regions based on a radial bias function kernel with a gamma of 0.2 and a C parameter of 10.0.

For the 10-fold cross-validation, stratified k-fold datasets were used. The relatively small size of the dataset limited the prospect for holdout validation, but k-fold cross validation estimated the accuracy of the modeling approach on new data. In this method, the dataset was randomly divided into k number of folds (i.e., 10 in this case) that were stratified, containing equal amounts of target classifiers. Then over k (i.e., 10) iterations, the data was trained on the k-1 (i.e., 9) data sets, using the remaining set for holdout validation. Then, k iterations are summarized with an overall accuracy score, estimating the overall accuracy of the model for predicting outcomes on new data.

## Data availability

All data is available upon reasonable request from the authors. The source data is attached as a Source Data file.

## Code availability

The python code for machine learning (Supplementary Code 1) and construction of Figs. 2–5 (Supplementary Code 2) is attached as a Supplementary Code file.

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

## Acknowledgements

The authors thank the Penn State Institute for CyberScience and the Penn State Materials Research Institute for providing access to Materials Studio 2019. R.T.M. thanks Dr. Jennifer Lynch for suggesting marine plastic debris as an alternative descriptor of ocean plastics.

## Author contributions

K.M. performed LogP(SA)$^{-1}$ analysis of plastics. J.D.C. contributed machine learning and edited the manuscript. R.T.M. maintained the database and wrote the manuscript.

## Competing interests

The authors declare no competing interests.
