## [Peer Review File · Nature Communications]

Reviewers' comments:

Reviewer #1 (Remarks to the Author):

The present papers is about prediction of plastic degradation in the oceans. This study is based on both computation of experimental data and computational prediction. The results are clearly presented and the classification of plastic according to persistency in the ocean is very informative. As an experimental scientist I do not understand in details the computational methods so I cannot discuss this aspect of the paper. I would have appreciated to understand a little bit better how the treatment of the existing experimental data is dealt with.

Paragraph Line 46-52 and paragraph Line 59-65 I think the order chosen to present first biotic degradation and then abiotic degradation is not related to the relative importance these two processes. It would be interesting to discuss the fact that biotic processes could initiate abiotic processes. How this could be incorporated in this computational approach? At least it should be mentioned in the paper.

Line 102 : how molecular entanglement is measured in polymer characterization? Is this information often provided? Could you add some bibliographic references for this type of characterization?

Figure 1: I understand the scheme presented in Fig 1 a) and b) for the probability of hydrolysis or photoinitiation. But I do not understand how the authors come up with the distinction between probability of biotic surface erosion or formation of microplastic? This is based on literature data? Could the authors specify the references used for this? Here surface erosion is opposed to microplastic formation, to my opinion both process are concomitant. Can the authors propose references to reinforce this representation?

Line 163 : How surface erosion is apprehended in the computation of experimental data ?

Line 164 : How do the authors integrate the autocatalytic hydrolysis of polylactic acid polymers?

Line 166: The authors present a classification for nylon degradation. Is this confirmed with experimental measurements? Can this be specified with a reference?

Line 169 and 170 : Indeed PE and PP are very represented at surface ocean. But it should be mentioned that there are also the polymers produced in the highest proportions.

There is anyway a difference between costal and open ocean for microplastic distribution. In coastal waters in addition to PE and PP other polymers are present whereas in open ocean PE and PP are the only ones detected within 1-5 mm.

There is another distribution regarding small microplastic (1-1000 μm). There is a wider distribution of polymers 1, 2. This is interesting, PP/PE proportion is lower within 1-5 mm compared to the range 1-1000 μm . Does that mean that PP is more susceptible to fragmentation/erosion than PE?

Line 175: Have the authors distinguished PDLA/PLLA? Is the distinction integrated in the crystallinity parameter?

Line 218: The authors are discussing here abiotic hydrolysis and biotic activity, they do not consider photodegradation?

Line 219 : This is not true for PE which Tg is below -100°C and which is very persistent. Can this aspect be discuss in the paper?

Figure 4 / figure 5 polyolefins are not localized in the same regions. Can this be discussed?

Line 339 The correlation with Tg values is not valid for PE

Line 345 - 346 I still do not understand how fragmentation and surface erosion can be placed in opposition. The authors should define erosion and fragmentation at the beginning of the paper and support these definitions with references

Line 351 The authors propose this computational approach to design degradable plastic. It should be important to underline that biodegradable plastic are interesting for specific and given applications because plastic recycling offers to reduce the carbon footprint of plastic compared to biodegradation 3

1. Enders, K.; Lenz, R.; Stedmon, C. A.; Nielsen, T. G. Abundance, size and polymer composition of marine microplastics $\geq 10 \mu\text{m}$ in the Atlantic Ocean and their modelled vertical distribution.

Mar Pollut Bull 2015, 100 (1), 70-81.

2. Ter Halle, A.; Jeanneau, L.; Martignac, M.; Jarde, E.; Pedrono, B.; Brach, L.; Gigault, J. Nanoplastic in the North Atlantic Subtropical Gyre. Environ Sci Technol 2017, 51 (23), 13689-13697.

3. Zheng, J. J.; Suh, S. Strategies to reduce the global carbon footprint of plastics (vol 9, pg 374, 2019). Nat Clim Change 2019, 9 (7), 567-567.

Reviewer #2 (Remarks to the Author):

The manuscript offered by Mathers and co-authors presents a data-driven approach to understanding the degradation (biotic and abiotic) of ocean plastics as it varies by polymer chemistry and other factors. While there has been a proliferation in studies recently on ocean plastics, rarely does one find any attention paid to the physical properties of different plastics and how that might differentiate them on the basis of their rates of degradation by any of the mechanisms available in different marine ecosystems. Accordingly, this manuscript carries with it an urgency that goes above and beyond the norm for the field. The use of machine learning here to identify the minimal number of physical descriptors to capture and predict the degradation behavior is particularly interesting, given that there is a clear connection between the data and the chemistry on outcomes. I highly recommend its publication in Nature Communications, provided the following comments are addressed.

Please re-do all Figures to avoid any that would otherwise have sloppily-drawn chemical structures, or where graphs and charts currently have blurry or have indistinct lines. The placement of panel designations a, b, c, etc. is weird and does not conform to author guidelines.

Make sure all letters used for variables are in italics, and the subscripts not.

The authors should quantify (i.e., include) the impact of density and surface energy on the degradation characteristics. Also, they should more clearly quantify the impact of various bond dissociation energies between the polymer classes for UV degradation / oxidation.

The authors might be more forthcoming with respect to their recommendations for how plastics should be manufactured and used, were we unable to control their leakage in the environment, based on their learning and analyses. In other words, is there a "safe" ocean plastic? Are all plastics ultimately poison to the environments in which they have leaked? Based on rates of leakage and degradation, how far in are we in the crisis? Etc.

Reviewer #3 (Remarks to the Author):

The authors compiled physical-chemical properties of a broad array of polymers (and plastics) as well as a data base of environmental degradation rates / processes that they extracted from the literature. The authors first present individual trends observed within their data set and then used machine learning to refine these trends.

Though this study has merits as the starting point of a critical review that can be used as a research starter to discuss the correlation between physical chemical properties and empirically observed degradation patterns, it falls far short of what it claims of being able to predict degradation trends of "oceanic" plastic, and even presents in my view a misguided characterization of the complex nature of marine plastic degradation. In particular, it ignores the role of anti-

oxidants which are the principle drivers of slow environmental degradation (anti-oxidants are added for the explicit purpose to slow the environmental degradation pathways the researchers discuss, as most polymers are unstable without antioxidants), and the study ignores plasticizers, which are added to alter the parameters the authors include are important, like glass transition temperature. Further, the assumed extrapolation of the different degradation rates applying to marine conditions, accounting for changes in e.g. temperature, wave action, was not addressed in detail and warrants further attention to warrant the title. From a pure statistical point of view, the model developed by in the study also lacks a validation to see how accurately it predicts data not included in the data set.

Overall, despite the many merits of the article, based on these concerns and the list of major comments below, I recommend this article for rejection in Nature Communications, but encourage the authors to develop their article more, as it has the potential to be a very useful resource in another journal, though with a more representative title like: "Trends for ranking environmental degradation rates of plastics based on physical properties and molecular structure".

Major comments:

1) No Statistical validation. The authors essentially used machine learning to fit the data they had, they did not use it to predict data not in their data set. Models like these should be tested (ideally blindly, or using a data set that is kept separate from the calibration data set), and the validation statistics presented. One classical way of doing this is to randomly pick 50% of the data, use this to calibrate the model using machine learning, and having it test the other 50%, and report the number of correct and incorrect predictions statistically. There are other ways to do this, such as including error bars to the predictions. Of course, the model becomes calibrated with less data, but the point is ultimately to check the validity of the underlying hypothetical assumptions.

2) Additives drive degradation rates in the environment. All commercial plastics have antioxidants. Without them, polymers degrade extremely rapidly, particularly through photo-oxidation. In any data set of degradation rates, it must be accounted for in context that the degradation method first degraded the anti-oxidants (mainly) and then the remaining polymers and other additives. All oceanic plastic would contain antioxidants. In principle, however, if some extra consideration was used to account for a range in delay of degradation under different environmental circumstances due to anti-oxidants, an approach like the authors use could be developed.

3) Plasticizers drive a lot of the important physical chemical properties of plastic. Softening agents and hardening agents change the properties that the authors found the most important, like crystallization, T_g. These are ignored. However, in a modified manuscript these could potentially be accounted for by introducing error bars on each of the parameters to account for the range of parameters allowed by use of different plasticizers.

4) Weathering also changes important physical chemical properties. There is a lot of active research now on how weathering of oceanic microplastic changes physical chemical properties (e.g. crystallinity), particularly at the surface. If these changes are important to degradation, as the authors conclude they are, then if weathering increases or decreases fragmentation rates should be accounted for too.

5) Semi-transparent database. What would be of extreme value to the community is the database the authors have compiled for this study, particular. However, only glimpses are seen (E.g. references used, visualization of correlations in the SI). For complete transparency it would be useful to have access to this data.

6) Quantitative vs Qualitative data presentation. Ultimately, this is a qualitative study that tries to find rules of thumb to rank trends in terms of what type of polymer degrades quicker than another type. Though this is of use, it would be informative to give quantitative descriptors of these rules of thumb (E.g. based on the data set, polyamides with T_g>X degrade quicker than T_g< x, 80% of the time). This could be combined with the validation work in point 1.

7) Correlation of parameters. The authors did not present explicitly how much the independent variables are correlated to each other, e.g. how well does T_g correlate with MW, crystallinity, etc. These are known from the polymer literature to correlate to some extent. More quantitative data could be given here too, referring to comment 6 and comment 1.

8) Conclusions (i.e. "the potential to inform the future design of plastics") If there was a need for rapidly degrading plastic, the technology is already there. The problem is finding the balance of having a commercially relevant polymer that is compatible with a waste reduction system (via degradation or recycling, which is mostly a logistic issue). Therefore I would stress to focus on environmental degradation prediction as the ultimate purpose of this work.

Minor comments

1) Missing oceanic weathering processes. The authors discuss photooxidation, hydrolysis and enzymatic biodegradation, lacking are mechanical forces. A recent trend in the literature is that coastal processes, particularly intense sunlight and crashing waves, are extremely important for fragmentation. Mechanical forces would lead to intense fragmentation of brittle plastics. (Chubarenko, I. P., Esiukova, E. E., Bagaev, A. V., Bagaeva, M. A., & Grave, A. N. (2018). Three-dimensional distribution of anthropogenic microparticles in the body of sandy beaches. *Science of the Total Environment*, 628, 1340-1351.)

2) Role of different oceanic temperatures, conditions in deep sea vs surface. This is something beyond the scope of the study, but an approach to address this could be mentioned.

3) Narrative voice. The manuscript deviates in voice from scientific level to anecdote story-telling, e.g. "L149 – "We wondered how many variables", "L229 – "which are discussed in sophomore organic chemistry" (which I also a bit condescending to the reader...)

4) L169. The disproportionate percentage" of low density plastics on the surface has also to do with relative emission rates of those floating plastics vs other ones, not just degradation rates.

5) Figure 2. The circles should ideally be ovals with the range of log P/Sa for the length of one axis and range of Enthalpies / crystallinities as the length of the other.

Dear Reviewers,

The revised manuscript has incorporated nearly all reviewer suggestions that were possible with the current capability of the database. As a result, this substantial revision addresses several serious limitations and clarifies a number of other issues. The highlighted text in yellow indicates changes/additions to the manuscript.

Response to Comments by Reviewer #1

- 1) Reviewer comment: "I would have appreciated to understand a little bit better how the treatment of existing experimental data is dealt with."

Author response: We agree and have added a paragraph in the method section to clarify how data on surface erosion was translated from numerical values to 3-tier and 5-tier categories.

Changes to manuscript: Calculation of surface erosion. Calculation of $\text{mg}/(\text{cm}^2 \text{ day})$ for sample films in the database was apprehended by dividing weight loss by surface area and the number of days that the films were exposed to seawater. We assume that $\text{mg}/(\text{cm}^2 \text{ day})$ values under biotic conditions probably have a small contribution from abiotic hydrolysis. In the case of PLA and PLLA,^{26,27,63} $\text{mg}/(\text{cm}^2 \text{ day})$ values are assumed to include the effect of mass loss from autocatalytic hydrolysis.

Categories. Surface erosion data in $\text{mg}/(\text{cm}^2 \text{ day})$ or biochemical oxygen demand (BOD) values in $\%/ \text{day}$ were converted to categories in the following manner. For 5-tier categories, very slow represented 0-2 % BOD/day and 0-0.0003 $\text{mg}/(\text{cm}^2 \text{ day})$, slow corresponded to 2-4 % BOD/day and 0.0003-0.003 $\text{mg}/(\text{cm}^2 \text{ day})$, medium represented 4-6 % BOD/day and 0.003-0.03 $\text{mg}/(\text{cm}^2 \text{ day})$, fast denoted 6-8 % BOD/day and 0.03-0.3 $\text{mg}/(\text{cm}^2 \text{ day})$, and very fast signified degradation processes > 8 % BOD/day and > 0.3 $\text{mg}/(\text{cm}^2 \text{ day})$. For 3-tier categories, slow denoted 0-4 % BOD/day and 0-0.003 $\text{mg}/(\text{cm}^2 \text{ day})$, medium represented 4-8 % BOD/day and 0.003-0.3 $\text{mg}/(\text{cm}^2 \text{ day})$, and fast signified >8 % BOD/day and > 0.3 $\text{mg}/(\text{cm}^2 \text{ day})$. Some $\text{mg}/(\text{cm}^2 \text{ day})$ values from laboratory experiments at higher temperatures were adjusted to fit ocean conditions. As a reference, poly(butylene adipate) (PBA dip) was ranked as medium for 5-tier and 3-tier categories.

- 2) Reviewer comment: "I think the order chosen to present first biotic degradation and then abiotic degradation is not related to the relative importance these two processes. It would be interesting to discuss the fact that biotic processes could initiate abiotic processes. How this could be incorporated in this computational approach. At least it should be mentioned in the paper."

Author response: These suggestions are extremely valid, but somewhat difficult to achieve with the current database as the current literature has very few journal articles that separate abiotic and biotic process. To address this issue, modification of the text surrounding Figure 2 hopefully clarifies that we separated the contribution for abiotic and biotic processes for as many polymers as possible and that biotic processes may help facilitate abiotic processes.

Changes to manuscript: Interestingly, while laboratory experiments for polyesters in Figure 2 fail to account for weathering processes and mechanical forces in the ocean, controlled conditions help separate the influence of abiotic hydrolysis from biodegradation and photo-initiated C-H bond oxidation. In scenarios where abiotic hydrolysis, biodegradation, and photo-initiated processes occur simultaneously, decreases in molecular weight via abiotic hydrolysis or photo-initiated reactions could facilitate biotic processes while enzymatic hydrolysis might promote abiotic hydrolysis.

- 3) Reviewer comment (Line 102): "How molecular entanglement is measured in polymer characterization? Is this information often provided? Could you add some bibliographic references for this type of characterization?"

Author response: Regarding entanglement molecular weight, we have decided to focus the manuscript on effect of molecular weight, glass transition temperature, crystallinity, enthalpy of melting, and LogP/SA values.

Changes to manuscript: Mention of entanglement molecular weight was removed and will be put into a follow up manuscript.

- 4) Reviewer comment: "Figure 1: I understand the scheme in Fig 1 a) and b) for the probability of hydrolysis or photoinitiation. But I do not understand how the authors come up with the distinction between probability of biotic surface erosion or formation of microplastic? This is based on literature data? Could the authors specify the references used for this? Here surface erosion is opposed to microplastic formation, to my opinion both process are concomitant. Can the authors propose references to reinforce this representation?"

Author response: We are glad to get feedback on the confusing aspects of Figure 1. Our attempt was to show that surface erosion and microplastic formation can both occur. To improve clarity, Figure 1 was changed to make our intent much clearer.

Changes to manuscript:

- a) Figure 1b arranges common types of plastics found in the ocean and a wide variety of other examples according to LogP/SA values. These initial efforts to investigate molecular structure indicate functional groups substantially lower the hydrophobicity relative to polyolefins. For example, Nylon 6 (LogP/SA = 0.0045 Å⁻²) and PCL (LogP/SA = 0.0096 Å⁻²) were considerably less than PE (LogP/SA =

0.0236 Å⁻²). Furthermore, this convenient method helped sort plastics into several groups.

- b) The third group ($\text{LogP/SA} > 0.015 \pm 0.002 \text{ \AA}^{-2}$) in Figure 1b corresponds to the most hydrophobic plastics that may not have functional groups for abiotic hydrolysis but most likely have a large percentage of C-H bonds susceptible to photodegradation. In addition to oxidation via photo-initiated processes, extremely slow surface erosion is observed for PE and PP.
- 5) Reviewer comment (Line 163): “How is surface erosion apprehended in the computation of experimental data?”

Author response: When possible, we calculated these values for each journal article in the database.

Changes to manuscript:

- a) Calculation of mg/(cm² day) for sample films in the database was apprehended by dividing weight loss by surface area and the number of days that the films were exposed to seawater.
- b) As denoted by the size of the circles in Figure 2, surface erosion was calculated using surface area of each plastic, mass loss, and number of days in seawater.
- 6) Reviewer comment (Line 164): “How do the authors integrate the autocatalytic hydrolysis of polylactic acid polymers?”

Author response: We assume that the effects of autocatalytic hydrolysis as well as abiotic hydrolysis and biotic hydrolysis are captured by the calculation of mg/(cm² day).

Changes to manuscript: In the case of PLA and PLLA,^{26,27,63} mg/(cm² day) values are assumed to include the effect of mass loss from autocatalytic hydrolysis.

- 7) Reviewer comment (Line 166): “The authors present a classification for nylon degradation. Is this confirmed with experimental measurements? Can this be specified with a reference?”

Author response: We had mentioned a key reference on nylon degradation several pages earlier, but have also mentioned this reference again.

Changes to manuscript: A similar trend was noted for nylons as proclivity to degrade decreased accordingly: Nylon 4 > Nylon 6 > Nylon 12.^{16,50}

- 8) Reviewer comment (Line 169 and 170): “Indeed PE and PP are very represented at surface ocean. But it should be mentioned that there are also the polymers produced in the highest proportions.”

Author response: We modified the text to reflect the reviewer’s suggestion about PE being produced in the highest proportion.

Changes to manuscript: “Recent studies confirm that plastics produced in the highest volume, like PE and PP, make up a disproportionate percentage of ocean plastics near the sea surface.⁵”

- 9) Reviewer comment: “There is anyway a difference between coastal and open ocean for microplastic distribution. In coastal waters in addition to PE and PP other polymers are present whereas in open ocean PE and PP are the only ones detected within 1-5 mm.

Author response: This is a challenging comment to address with the current database. We appreciate this question and wish to express our goal focuses on predicting degradation trends via analysis of physical properties and surface erosion rather than predicting the location of microplastics in the ocean. In order to clarify the intent of our data analysis, a section on relevance was added. This includes a discussion of bacteria estimates in coastal areas and open ocean (Earth Syst. Sci. Data, 4, 101–106, 2012). Consequently, we think the models in Figures 4 and 5 can apply to both coastal and open ocean locations.

Changes to manuscript: Relevance. In order to understand the applicability of trends in Figures 4 and 5, data analysis involved the following considerations: First, to cover a wide range of environmental conditions, data collection included temperatures ranging from $\sim 0\text{ }^{\circ}\text{C}$ ⁵¹ to $> 30\text{ }^{\circ}\text{C}$,^{32,52} shallow ocean depths (1-10 m),^{20,53} deep seawater ($\sim 300\text{ m}$ to $> 600\text{ m}$),¹⁵ and simulated deep sea pressure.⁵⁴ Second, this study focuses on plastics in direct contact with either real seawater or artificial seawater. As a result, certain scenarios, like microplastics that wash onto beaches due to mechanical action of waves and weather on beaches in the dry state or cycle back and forth between the ocean and beach, exceeds the limit of the current database. Third, regarding the presence of bacteria that could potentially result in biodegradation, coastal regions and open oceans to a depth of 225 m have similar number of cells.⁵⁵ Consequently, we assume the degradation trends in Figures 4 and 5 could apply to both coastal and open ocean.

- 10) Reviewer comment: “There is another distribution regarding small microplastic (1-1000 μm). There is a wider distribution of polymers 1, 2. This is interesting, PP/PE proportion is lower within 1-5 mm compared to the range of 1-1000 μm . Does this mean that PP is more susceptible to fragmentation/erosion than PE?”

Author response: The text was modified to mention small microplastic (1-1000 μm) along with appropriate references. Regarding the susceptibility of PP fragmentation/erosion, we also added a sentence to clarify.

Changes to manuscript:

- a) “Due to a wide variety of environmental factors, such as exposure to UV radiation, wind, waves, seawater, and bacteria, plastic waste experiences concurrent influences leading to cracking, surface erosion, abrasion, and breakdown to mesoplastic ($\sim 5\text{-}20\text{ mm}$), large microplastic ($\sim 1\text{-}5\text{ mm}$), small microplastic ($\sim 20\text{-}999\text{ }\mu\text{m}$), and nanoplastic ($< 1\text{ }\mu\text{m}$) sized pieces.⁷⁻⁹”
- b) However, some plastics without functional groups, such as polyolefins, exhibit very slow degradation even though T_g values are quite low. Furthermore, additives in commercial polyolefins slow degradation for polyethylene (0.45 wt. %/month) and polypropylene (0.39 wt. %/month).²⁰

- 11) Reviewer comment (Line 175): “Have the authors distinguished PDLA/PLLA? Is the distinction integrated in the crystallinity parameter?”

Author response: The database does contain PLA and PLLA. These polyesters differentiate mainly by the amount of crystallinity.

Changes to manuscript:

- a) While the ranking in Figure 1 generally correlates with proclivity for polyester degradation, plastics with T_g values $>$ ocean temperature, like PLA, PLLA, and PET, degrade more slowly than expected.^{26,27}
 - b) A comparison of PLA and PLLA (see Supplementary Data 4) indicate the increased % crystallinity of PLLA slows surface erosion.
 - c) In Supplementary Data 4, we included two graphs of surface erosion versus % crystallinity and surface erosion versus T_m values.
- 12) Reviewer comment (Line 218): “The authors are discussing here abiotic hydrolysis and biotic activity, they do not consider photodegradation?”

Author response: Relating to photodegradation, we modified the text to clarify photo-initiated C-H bond oxidation could occur with abiotic hydrolysis and biodegradation.

Change to manuscript: Interestingly, while laboratory experiments for polyesters in Figure 2 fail to account for weathering processes and mechanical forces in the ocean, controlled conditions help separate the influence of abiotic hydrolysis from biodegradation and photo-initiated C-H bond oxidation.

- 13) Reviewer comment (Line 219): “This is not true for PE which T_g is below -100°C and which is very persistent. Can this aspect be discussed in the paper?”

Author response: The statement about polymers with T_g values below ocean temperature was clarified and rewritten.

Changes to manuscript: Figure 3 investigated the combined effect of T_g and molecular weight on degradation. In the context of photo-initiated C-H bond oxidation, abiotic hydrolysis, and biotic activity, several observations emerged. First, degradation trends paralleled T_g values and decreased accordingly: linear polymers (i.e. PCL) $>$ branched polymers with methyl groups (i.e. PHB and PHBV) $>$ polymers with cyclic rings and functional groups in polymer chain (i.e. PBAT, PET, PC) $>$ polymers with cyclic rings and all carbon atoms in polymer chain [i.e. PS, poly(vinyl pyrrolidone)]. These trends suggest degradation occurs more rapidly with T_g values below ocean temperature.

- 14) Reviewer comment: “Figure 4 / figure 5 polyolefins are not localized in the same regions. Can this be discussed?”

Author response: Great point concerning the location of polyolefins in Figures 4 and 5. We added a sentence to the discussion of Figure 4.

Changes to manuscript: In the case of Figure 4, most errors resulted for plastics, like polyolefins, when LogP/SA exceeded -0.010 \AA^{-2} or when $\text{LogP}/\text{SA} < 0$ as demonstrated by water soluble polymers, like PVA. However, moving from the 2-

level tree in Figure 4 to the 3-level tree, the number of inaccurate predictions decreased from 15 to 8, respectively. This suggests Figure 4 applies to a narrower window of LogP/SA values than Figure 5.

- 15) Reviewer comment (Line 339): “The correlation with T_g values is not valid for PE”

Author response: To clarify the correlation between T_g and degradation, the text was changed to:

Changes to manuscript: Third, in Figure 3, degradation of plastics with heteroatoms (i.e. O, N) substantially slowed as T_g values increased above ocean temperature. For example, slower degradation was noted for PLA (T_g ~ 65 °C) and PET (T_g ~ 78 °C) compared to PBAT (T_g ~ -36 °C).

- 16) Reviewer comment: “I still do not understand how fragmentation and surface erosion can be placed in opposition. The authors should define erosion and fragmentation at the beginning of the paper and support these definitions with references”

Author response: We thank the reviewer for mentioning the lack of clarity regarding fragmentation and surface erosion in lines 345-346. We intended to convey that both surface erosion and fragmentation occur simultaneously on different timescales. To avoid confusion, we changed the manuscript.

Changes to manuscript:

- a) Surface erosion refers to mass loss over time for a given surface area in units of mg/(cm² day).^{26,37}
- b) First, in Figure 1, functional groups, like carbonates, esters, and amides, lower the magnitude of LogP/SA values relative to PE and facilitate abiotic hydrolysis and biotic pathways for degradation. In contrast, larger LogP/SA values (i.e. > 0.015 Å⁻²) indicate a substantial fraction of C-H bonds in the polymer structure. Nonetheless, even though photo-initiated C-H bond oxidation is feasible, the presence of additives (i.e. antioxidants, light stabilizers)²³ will delay degradation.

- 17) Reviewer comment (Line 351): “The authors propose this computational approach to design degradable plastic. It should be important to underline that biodegradable plastic are interesting for specific and given applications because plastic recycling offers to reduce the carbon footprint of plastic compared to biodegradation 3”

Author response: We agree that plastic recycling is a better strategy than simply relying on biodegradation. The text was modified.

Changes to manuscript: While efforts at sustainable development of plastics has increased in the last decade,⁶⁹⁻⁷¹ these efforts struggle to meet the growing challenge of plastic waste.⁷² Consequently, we emphasize the need for recycling to reduce the global carbon footprint and highlight strategies aimed to accelerate degradation.⁷²

- 18) The suggested reference by Enders et al. and Ter Halle et al. were added and fit nicely into the introduction. The reference by Zheng et al. was added to the conclusion.

Changes to manuscript:

- This multi-faceted situation has led to questions about size distribution and composition along with inquiries into how bulk properties of plastics change in the ocean.⁹⁻¹¹
- While efforts at sustainable development of plastics has increased in the last decade,⁶⁹⁻⁷¹ these efforts struggle to meet the growing challenge of plastic waste.⁷²

Reviewer #2

- Reviewer comment: "Please re-do all Figures to avoid any that would otherwise have sloppily-drawn chemical structures, or where graphs and charts currently have blurry or have indistinct lines. The placement of panel designations a, b, c, etc. is weird and does not conform to author guidelines."

Author response: Thanks for the feedback. As suggested, we revised Figures 1-5 to conform to author guidelines.

Changes to Figures:

a

b

2) Reviewer comment: “Make sure all letters used for variables are in italics, and the subscripts not.”

Author comment: The issue with italicized variables (i.e. T_g , M_n , LogP/SA) was fixed throughout the manuscript.

For example:

- a) Figure 3 investigated the combined effect of T_g and molecular weight on degradation. In the context of photo-initiated C-H bond oxidation, abiotic hydrolysis, and biotic activity, several observations emerged. First, degradation trends paralleled T_g values and decreased accordingly: linear polymers (i.e. PCL) > branched polymers with methyl groups (i.e. PHB and PHBV) > polymers with cyclic rings and functional groups in polymer chain (i.e. PBAT, PET, PC) > polymers with cyclic rings and all carbon atoms in polymer chain [i.e. PS, poly(vinyl pyrrolidone)]. These trends suggest degradation occurs more rapidly with T_g values below ocean temperature. However, some plastics without functional groups, such as polyolefins,

exhibit very slow degradation even though T_g values are quite low. Furthermore, additives in commercial polyolefins slow degradation for polyethylene (0.45 wt. %/month) and polypropylene (0.39 wt. %/month).²⁰ Second, the fastest abiotic hydrolysis occurred for molecular weights below ~25 kg/mol. However, when $T_g <$ ocean temperature, enzymatic activity degraded PHB ($T_g \sim 2-5$ °C) reasonably well even when molecular weight was 200k-700k g/mol. Third, Figure 3 provides a framework for estimating plastics based on two common experimental measurements, namely molecular weight and T_g .

- b) Fourth, a comparison of Figures 3-5 suggest a hierarchy of features for predicting the likelihood of degradation. In this regard, LogP/SA , which reflects composition, may be more useful than molecular weight when comparing water-soluble and water insoluble polymers. Furthermore, LogP/SA , molecular weight, and T_g apply to both amorphous and partially crystalline (i.e. semicrystalline) plastics whereas Figure 2 focused on parameters that relate to semicrystalline polymers (i.e. % crystallinity and enthalpy of melting).
 - c) Bulk property descriptors included density, weight-average molecular weight (M_w), number-average molecular weight (M_n), dispersity (M_w/M_n), glass transition temperature (T_g), melting temperature (T_m), % crystallinity, and enthalpy of melting (i.e. amount of energy required).
 - d) Although % crystallinity, enthalpy of melting and T_m values are all informative, crystallinity and enthalpy of melting allow an easier comparison of polyesters and polyamides (i.e. Nylon 6, Nylon 6,6) than T_m values. For example, the relationship between T_m values and degradation show opposite trends for polyesters and polyamides. As such, degradation decreases as follows: Nylon 4 ($T_m \sim 267$ °C) > Nylon 6,6 ($T_m \sim 264$ °C) > Nylon 6 ($T_m \sim 220$ °C).^{16,50} In contrast, polyesters with lower T_m values, such as PCL ($T_m \sim 60$ °C), show faster degradation than poly(ethylene succinate) (PES) ($T_m \sim 104$ °C).
- 3) Reviewer comment: "The authors should quantify (i.e., include) the impact of density and surface energy on the degradation characteristics."

Author response: Regarding the impact of density and surface energy, we think this is a good idea. In the manuscript, density was briefly discussed in the text and correlates with LogP/SA values (Supplementary Data 2 and 3). Additionally, more information on density has been provided in Supplementary Data 12. Mention of surface energy and two additional references were also added to the introduction.

Changes to manuscript:

- a) For density determination (used to calculate mass for surface erosion), calibration curves (see Supplementary Data 12) for polyesters, nylons, and PHBV were created.
- b) First, depending on surface energy, bacteria colonize surfaces in the ocean and have a propensity for biofilm formation.¹³⁻¹⁵

- 4) Reviewer comment: “Also, they should more clearly quantify the impact of various bond dissociation energies between the polymer classes for UV degradation / oxidation.”

Author response: This is a good idea to expand our discussion of photo-degradation. We have added some more references and changed the text. In addition, we added a sentence about bond dissociation energies.

Changes to manuscript:

- a) As a result, most commercial plastics contain additives such as antioxidants and light stabilizers that delay degradation.^{24,25} In general terms, polymers without tertiary hydrogens, like poly(methyl methacrylate) (PMMA) and polytetrafluoroethylene (PTFE), are often highly stable. In contrast, others are moderately stable (PET, PC) or poorly stable due to the presence of tertiary C-H bonds [PS, poly(vinyl chloride) (PVC)], allylic C-H bonds [polyisoprene (PI), polybutadiene (PBD)], C=O bonds [polyamides (PA), polyurethanes (PU)], and catalyst residues (PE, polypropylene (PP)).²⁴
- b) Overall, a comparison of plastics with and without tertiary C-H bonds reveals reactivity (i.e. bond dissociation energies) decreases as follows: PVC > PS > PP > PE.⁴
- 5) Reviewer comment: “The authors might be more forthcoming with respect to their recommendations for how plastics should be manufactured and used, were we unable to control their leakage in the environment, based on their learning and analyses. In other words, is there a “safe” ocean plastic? Are all plastics ultimately poison to the environments in which they have leaked? Based on rates of leakage and degradation, how far in are we in the crisis? Etc.”

Author response: Great questions! In view of Reviewer #3's suggestion that we focus the manuscript on environment degradation, the mention of “future design” was replaced with an emphasis on quantifying degradation (i.e. equations 2 and 3). We hope to consider synthetic recommendations in a follow up manuscript.

Changes to manuscript:

- a) Even with two features, like molecular weight versus T_g or LogP/SA versus T_g , the compelling results underscore the connection between environmental degradation and structure-property relationships.
- b) Moving forward, we propose data-driven, machine learning techniques, such as the classification trees in Figures 4-5, inform predictive models like equations 2 and 3 by identifying the important physical property parameters.

Reviewer #3

We thank the reviewer for a thorough and detailed evaluation of this manuscript. The reviewer's comments were very helpful during our revision. We concur with the importance of the reviewer's comprehensive list. Furthermore, we agree this list (that includes anti-

oxidants, plasticizers, temperature, wave action, mechanical forces, the issue of deep sea vs surface, and weathering of plastics) contributes to what the reviewer called “the complex nature of marine plastics.” During our analysis of data, we have considered some issues in detail (i.e. physical properties, temperature, surface area), discussed some issues sparingly (i.e. additives), and omitted other issues (i.e. wave action and mechanical forces). However, the revised manuscript includes significant changes to address these limitations.

- 1) Reviewer’s Major comment #1: “No Statistical validation. The authors essentially used machine learning to fit the data they had, they did not use it to predict data not in their data set. Models like these should be tested (ideally blindly, or using a data set that is kept separate from the calibration data set), and the validation statistics presented. One classical way of doing this is to randomly pick 50% of the data, use this to calibrate the model using machine learning, and having it test the other 50%, and report the number of correct and incorrect predictions statistically. There are other ways to do this, such as including error bars to the predictions. Of course, the model becomes calibrated with less data, but the point is ultimately to check the validity of the underlying hypothetical assumptions.”

Author response: Thanks for mentioning the importance of statistical validation. We had previously conducted statistical validation but not discussed the results in the first submission of the manuscript. However, we agree that statistical validation is an important aspect of machine learning efforts and have included a discussion of validation in the revised version of the manuscript. To validate the data, we have used a k-fold cross validation (CV) strategy. This strategy works well for the size of the dataset and provides a relative confidence as to where the models stand at this time, which ranged from 57-63%.

Changes to manuscript:

- a) Accuracy of the decision tree model on the training data (see Supplementary Data 5) increased from 75.4% with 2-levels containing 2 features to 86.9% with 3-levels comprising 4 features. Due to the relatively limited number of samples, applying ten-fold cross-validation to the models provided an accuracy of the 57.8% for the 2-level model and 63.2% for the 3-level model.
 - b) For the 10-fold cross-validation, stratified k-fold datasets were used. The relatively small size of the dataset limited the prospect for holdout validation, but k-fold cross validation estimated the accuracy of the modeling approach on new data. In this method, the dataset was randomly divided into k number of folds (i.e. 10 in this case) that were stratified, containing equal amounts of target classifiers. Then over k (i.e. 10) iterations, the data was trained on the k-1 (i.e. 9) data sets, using the remaining set for holdout validation. Then, k iterations are summarized with an overall accuracy score, estimating the overall accuracy of the model for predicting outcomes on new data.
- 2) Reviewer’s Major comment #2: “Additives drive degradation rates in the environment. All commercial plastics have antioxidants. Without them, polymers degrade extremely rapidly, particularly through photo-oxidation. In any data set of degradation rates, it must be accounted for in context that the degradation method

first degraded the anti-oxidants (mainly) and then the remaining polymers and other additives. All oceanic plastic would contain antioxidants. In principle, however, if some extra consideration was used to account for a range in delay of degradation under different environmental circumstances due to anti-oxidants, an approach like the authors use could be developed.”

Author response: We completely agree that commercial plastics contain additives and these additives slow degradation rates. To put our study in context, a sentence was added to convey that the database had a substantial number of commercial plastics. Although the introduction briefly mentioned the issue of additives, we agree more discussion would be very helpful. As a result, more explanation was added to the introduction. We really like the reviewer’s suggestion about the “range in delay of degradation.” Unfortunately, the current amount of literature data on this topic is limited and insufficient to model or draw conclusions. We hope to address this issue in when more experimental data becomes available.

Changes to manuscript:

- a) Plastics in the database included commercial samples (69) and those made in a laboratory (46).
 - b) However, some plastics without functional groups, such as polyolefins, exhibit very slow degradation even though T_g values are quite low. Furthermore, additives in commercial polyolefins slow degradation for polyethylene (0.45 wt. %/month) and polypropylene (0.39 wt. %/month).²⁰
 - c) In contrast, larger LogP/SA values (i.e. $> 0.015 \text{ \AA}^{-2}$) indicate a substantial fraction of C-H bonds in the polymer structure. Nonetheless, even though photo-initiated C-H bond oxidation is feasible, the presence of additives (i.e. antioxidants, light stabilizers)²³ will delay degradation.
- 3) Reviewer’s Major comment #3: “Plasticizers drive a lot of the important physical chemical properties of plastic. Softening agents and hardening agents change the properties that the authors found the most important, like crystallization, T_g . These are ignored. However, in a modified manuscript these could potentially be accounted for by introducing error bars on each of the parameters to account for the range of parameters allowed by use of different plasticizers.”

Author response: We recognize the importance of plasticizers and think T_g is a great way to distinguish between polymers with and without plasticizers. While it is true that some polymers, like PVC, contain large amounts of plasticizer, many polymers contain very small amounts of plasticizer. In addition, based on very recent literature searches, no journal articles are available which compare the degradation of plastics with and without plasticizer under similar environmental conditions. To clarify, a comment was added to the section on relevance. As such, the data does not ignore the issue of plasticizers, but the text is vague and needs to be clarified.

Changes to manuscript:

- a) Fourth, since T_g values exhibit sensitivity to polymer structure,⁵⁶ molecular weight (i.e. Flory-Fox equation),⁵⁷ crosslinking,⁵⁸ and plasticizers,⁵⁹ this metric has some comprehensive potential. We expect differences in heating rates during T_g

measurements as well as small quantities of plasticizer introduces variability in the data, but assume this error is nominal compared to the breath of the categories in Figures 4 and 5. Furthermore, due to the availability of data, Figures 4 and 5 includes commercial samples of virgin PVC.³⁹

- 4) Reviewer's Major comment #4: "Weathering also changes important physical chemical properties. There is a lot of active research now on how weathering of oceanic microplastic changes physical chemical properties (e.g. crystallinity), particularly at the surface. If these changes are important to degradation, as the authors conclude they are, than if weathering increases or decreases fragmentation rates should be accounted for too."

Author response: Thanks for commenting about the important role of weathering. We created a new paragraph entitled "Environmental conditions." This section discusses the relationship between seawater temperature and surface erosion. Three new graphs (Supplementary Data 7-9) were added and these form the basis for equation 2 which describes the parameters which influence the rate of surface erosion (k). Then, equation 3 serves as a starting point for preliminary discussion of mechanical forces. In addition, during discussion of Figure 5, more text was added.

Changes to manuscript:

- a) The relationship between surface erosion rates (k) and physical properties in equation 2 models data from polyesters and polyamides with $\text{LogP/SA} > 0$ and enthalpy of melting $< 85 \text{ J/g}$. Essentially, equation 2 depends on $(T_{\text{water}} - T_g)/(\text{LogP/SA})$ and predicts the slope of surface erosion versus temperature with units of $\text{mg}/(\text{cm}^2 \text{ day } ^\circ\text{C})$. We hypothesize predictions extend to other polymers containing functional groups with carbonyls (C=O), such as PC and PU. To test this hypothesis, predictions for PC, PU, and PET (see Supplementary Data 10) seem reasonable compared to PCL, PHBV, Nylon 6, and PLA. As a caveat, the intent of equation 2 focuses on amorphous or semi-crystalline polymers (enthalpy of melting $< \sim 90 \text{ J/g}$) with $\text{LogP/SA} > 0$. Outside of these parameters, equation 2 overestimates k for certain polyesters with larger enthalpy of melting values, such as PBS ($\sim 132 \text{ J/g}$)⁶² or PBSeb ($\sim 125 \text{ J/g}$).²⁵

$$\text{rate of surface erosion } (k) = \exp\left(\left(\frac{T_{\text{water}} - T_g}{\frac{\text{LogP}}{\text{SA}}}\right) - 28795\right) / 4177.3 \quad (2)$$

- b) In a preliminary effort to expand upon equation 2 and capture the multi-faceted processes that influence degradation, we propose a simple model in equation 3. Inspired by efforts to describe weathering,^{61,67,68} this model assumes the total amount of erosion (E_{total}) depends on abiotic processes, biotic processes, seawater temperature (T_{water}), and mechanical forces (E_{waves}). As such, k from equation 2 describes the rate of abiotic and biotic processes and b is the y-intercept in the absence of mechanical forces. To calculate E_{waves} , the difference in surface erosion between ocean conditions and sheltered locations is proposed. For example, surface erosion of PHBV increased when exposed to coastal locations⁵⁴ [$E_{\text{waves}} = 0.017 \text{ mg}/(\text{cm}^2 \text{ day})$] and an estuary⁶⁹ [$E_{\text{waves}} = 0.005 \text{ mg}/(\text{cm}^2 \text{ day})$]

compared to sheltered mangroves. Although more literature data is needed to further explore the limitations of equation 3, initial data analysis (see Supplementary Data 11) serves as a starting point for future discussions.

$$E_{\text{total}} = kT_{\text{water}} + b + E_{\text{waves}} \quad (3)$$

- 5) Reviewer's Major comment #5: "Semi-transparent database. What would be of extreme value to the community is the database the authors have compiled for this study, particular. However, only glimpses are seen (E.g. references used, visualization of correlations in the SI). For complete transparency it would be useful to have access to this data."

Author response: We look forward to sharing this data and include the calculated parameters and literature data for Figures 1-5 in Supplementary Data 11.

Changes to manuscript: The data for Figures 1-5 is shown in Supplementary Data 13.

- 6) Reviewer's Major comment #6: "Quantitative vs Qualitative data presentation. Ultimately, this is a qualitative study that tries to find rules of thumb to rank trends in terms of what type of polymer degrades quicker than another type. Though this is of use, it would be informative to give quantitative descriptors of these rules of thumb (E.g. based on the data set, polyamides with $T_g > X$ degrade quicker than $T_g < X$, 80% of the time). This could be combined with the validation work in point 1."

Author response: As mentioned during the discussion of weathering, we have added equation 2 to determine rate (k) as a function of temperature and equation 3 to quantify the effect of mechanical forces.

Changes to manuscript:

a)
$$\text{rate of surface erosion } (k) = \exp\left(\left(\frac{T_{\text{water}} - T_g}{\frac{\text{LogP}}{\text{SA}}}\right) - 28795\right) / 4177.3 \quad (2)$$

b)
$$E_{\text{total}} = kT_{\text{water}} + b + E_{\text{waves}} \quad (3)$$

- 7) Reviewer's Major comment #7: "Correlation of parameters. The authors did not present explicitly how much the independent variables are correlated to each other, e.g. how well does T_g correlate with MW, crystallinity, etc. These are known from the polymer literature to correlate to some extent. More quantitative data could be given here too, referring to comment 6 and comment 1."

Author response: We appreciate the question about correlation of parameters. Interestingly, this is a good reason to have T_g on the x-axis of Figures 4 and 5 and allows these figures to have some underlying dependence on molecular weight and molecular structure. Overall, many features (variables) were examined in this study, and after identifying the most important features, an in-depth correlation of features analysis appeared to be largely unnecessary. We did provide a correlation matrix of the most relevant features in the Supplemental Data 2. Additionally, we added a note about some of the highly correlated features that allowed for feature reduction during machine learning.

Changes to manuscript:

- a) Fourth, since T_g values exhibit sensitivity to polymer structure,⁵⁷ molecular weight (i.e. Flory-Fox equation),⁵⁸ crosslinking,⁵⁹ and plasticizers,⁶⁰ this metric has some comprehensive potential.
 - b) A second group in Figure 1b comprises insoluble plastics ($0 < \text{LogP/SA} < 0.013 \pm 0.002 \text{ \AA}^{-2}$) susceptible to surface erosion via biodegradation and abiotic hydrolysis through exposure to seawater. Within this category, the propensity for polyester surface erosion correlates with hydrophobicity when the T_g values $<$ ocean temperature.
 - c) As a result of the correlation between LogP/SA and other features (Supplementary Data 3), the list of seven possible predictors shortened to five: molecular weight, T_g , % crystallinity, enthalpy of melting, and LogP/SA .
 - d) Even with two features, like molecular weight versus T_g or LogP/SA versus T_g , the compelling results underscore the connection between environmental degradation and structure-property relationships.
- 8) Reviewer's Major comment #8: "Conclusions (i.e. "the potential to inform the future design of plastics") If there was a need for rapidly degrading plastic, the technology is already there. The problem is finding the balance of having a commercially relevant polymer that is compatible with a waste reduction system (via degradation or recycling, which is mostly a logistic issue). Therefore I would stress to focus on environmental degradation prediction as the ultimate purpose of this work."

Author response: We like the suggestion about focusing the manuscript on environment degradation. In the text, the mention of "future design" was removed.

Changes to manuscript:

- a) Title of the manuscript: Trends for Ranking Environmental Degradation of Plastic Marine Debris Based on Physical Properties and Molecular Structure
 - b) Moving forward, we propose data-driven, machine learning techniques, such as the classification trees in Figures 4-5, inform predictive models like equation 2 by identifying the important physical property parameters.
 - c) Even with two features, like molecular weight versus T_g or LogP/SA versus T_g , the compelling results underscore the connection between environmental degradation and structure-property relationships.
- 9) Minor comment #1: "Missing oceanic weathering processes. The authors discuss photooxidation, hydrolysis and enzymatic biodegradation, lacking are mechanical forces. A recent trend in the literature is that coastal processes, particularly intense sunlight and crashing waves, are extremely important for fragmentation. Mechanical forces would lead to intense fragmentation of brittle plastics. (Chubarenko, I. P., Esiukova, E. E., Bagaev, A. V., Bagaeva, M. A., & Grave, A. N. (2018). Three-dimensional distribution of anthropogenic microparticles in the body of sandy beaches. *Science of the Total Environment*, 628, 1340-1351.)"

Author response: We added equation 3 to model available data. The purpose of Figure 2 was clarified. The suggested reference was also added.

Changes to manuscript:

- a) In a preliminary effort to expand upon equation 2 and capture the multi-faceted processes that influence degradation, we propose a simple model in equation 3. Inspired by efforts to describe weathering,^{61,67,68} this model assumes the total amount of erosion (E_{total}) depends on abiotic processes, biotic processes, seawater temperature (T_{water}), and mechanical forces (E_{waves}).
 - b) Interestingly, while laboratory experiments for polyesters in Figure 2 fail to account for weathering processes and mechanical forces in the ocean, controlled conditions help separate the influence of abiotic hydrolysis from biodegradation and photo-initiated C-H bond oxidation.
 - c) The influence of weathering on plastic debris represents a complex issue that depends on a number of parameters, like sample depth, temperature, mechanical forces, and sunlight.^{10,22,61}
- 10) Reviewer's Minor comment #2: "Role of different oceanic temperatures, conditions in deep sea vs surface. This is something beyond the scope of the study, but an approach to address this could be mentioned."

Author response: We appreciate the suggestion of ocean temperature and have discussed this aspect in the paragraphs on relevance and environmental conditions.

Changes to manuscript:

- a) In order to understand the applicability of trends in Figures 4 and 5, data analysis involved the following considerations: First, to cover a wide range of environmental conditions, data collection included temperatures ranging from ~ 0 °C⁵² to > 30 °C,^{32,53} shallow ocean depths (1-10 m),^{20,54} deep seawater (~ 300 m to > 600 m),¹⁵ and simulated deep sea pressure.⁵⁵
 - b) Graphs which data for surface erosion versus temperature were added to Supplementary Data 7 and 8. These graphs contain all the available depth data.
- 11) Reviewer's Minor comment #3: "Narrative voice. The manuscript deviates in voice from scientific level to anecdote story-telling, e.g. "L 149 – "We wondered how many variables", "L 229 – "which are discussed in sophomore organic chemistry" (which I also a bit condescending to the reader...)"

Author response: Regarding the style of writing, we removed the comment on organic chemistry.

- 12) Reviewer's Minor comment #4: "L 169. The disproportionate percentage of low density plastics on the surface has also to do with relative emission rates of those floating plastics vs other ones, not just degradation rates."

Author response: To clarify the reviewers comment on the "disproportionate percentage of low density plastics on the surface," we modified the text.

Changes to manuscript: Recent studies confirm that plastics produced in the highest volume, like PE and PP, make up a disproportionate percentage of ocean plastics near the sea surface.⁵

REVIEWERS' COMMENTS:

Reviewer #1 (Remarks to the Author):

Thank you for the discussion

Reviewer #2 (Remarks to the Author):

The authors have thoroughly addressed the major comments made by the reviewers. I recommend that this manuscript be published.

The Figures still need to be re-worked such that the variables used in the figures and plots conform to the author guidelines. variables should be in italics, with their subscripts not in italics.

Reviewer #3 (Remarks to the Author):

After reading the replies to the reviewers and the revised manuscript, my opinion of this work has increased substantially. The authors, addressed my major concerns in a satisfactory manner, such as including a statistic validation of their model, at least mentioning the important role of additives on degradation (via affecting stability or crystallinity), adding a completely new section to account for the possible sources of weathering process, discussing the correlation of the parameters, and adjusting phrasing of the title. Throughout the paper, also in response to the other reviewers, the authors added a large amount of much needed mechanistic detail. I feel that now this revised contribution has reached a lack of completion that it will be useful to the (micro)plastic community and readers of Nature Communications.

Very minor comments that do not require re-review:

- Line 28. Phrasing. Last sentence of the abstract starting with "Then, a ..." does not read smoothly" consider rephrasing

- Figure 3. Are all polymers in the legend on this chart? For instance, I could not find the important polyolefin nor PVC; are they buried under another dots? Are they needed in the legend?

- Figure 3d. Why are there data with T_g less than 0 degrees here. Consider another cutoff, like T_g < -5C

- The texts still contains a lot of "we"s , though fewer than last time; I think the remaining cases could be removed, e.g. with passive voice.

Response to Comments by Reviewer #1

1) Reviewer comment: "Thank you for the discussion."

Author response: Many thanks for carefully reading this manuscript.

Response to Comments by Reviewer #2

1) Reviewer comment: "The Figures still need to be re-worked such that the variables used in the figures and plots conform to the author guidelines. variables should be in italics, with their subscripts not in italics."

Author response: We had previously attempted to fix this problem, but had some confusion about the best way to represent $\text{Log}P(\text{SA})^{-1}$. After communication with an editor, we clearly understand the guidelines.

Changes to Figure 1:

Changes to Figure 2:

Changes to Figure 3:

Changes to Figure 4:

Changes to Figure 5:

Response to Comments by Reviewer #3

- 1) Reviewer comment: "Line 28. Phrasing. Last sentence of the abstract starting with "Then, a ..." does not read smoothly" consider rephrasing"

Author response: We changed the text.

Changes to manuscript: Then, Figure 3 involves a data-analytics approach to evaluate a wider range of experimental conditions (i.e. laboratory, ocean) using 5-tier categories.

- 2) Reviewer comment: "Figure 3. Are all polymers in the legend on this chart? For instance, I could not find the important polyolefin nor PVC; are they buried under another dots? Are they needed in the legend?"

Author response: Thanks for mentioning.

Changes to Figure 3: We modified the legend and removed PVC and polyolefin.

- 3) Reviewer comment: “Figure 3d. Why are there data with T_g less than 0 degrees here. Consider another cutoff, like T_g < -5C”

Author response: We intended to show how Figure 3b and 3d overlapped.

- 4) Reviewer comment: “The texts still contains a lot of "we"s , though fewer than last time; I think the remaining cases could be removed, e.g. with passive voice.”

Author response: We agree and have modified the manuscript to eliminate 8 instances involving the word “we.”

Changes to manuscript:

- To elaborate, **the data contains** instances were differences in environmental conditions as well as comparison of commercial materials with those produced in a laboratory produced a range of degradation behavior.
- Since the 3-level tree in Figure 5 produced less incorrect predictions than molecular weight and T_g (see Supplementary **Figure 6**), **the location** of PET in the slow category of Figure 5b is more appropriate for commercial plastics than the medium category in Figures 3c and 4b.
- Consequently, degradation** trends in Figures 4 and 5 could apply to both coastal and open ocean.
- As a result, the complexity of analysis** systematically increased in Figures 1-5.
- Differences in heating rates during T_g measurements as well as small quantities of plasticizer introduces variability in the data, but this error is nominal** compared to the breath of the categories in Figures 4 and 5.

- f) In a preliminary effort to expand upon equation 2 and capture the multi-faceted processes that influence degradation, a simple model in equation 3 is proposed.
- g) As a result, these inevitable boundary errors differ from conflicting literature data.
- h) The $\text{mg cm}^{-2} \text{ day}^{-1}$ values under biotic conditions are assumed to have a small contribution from abiotic hydrolysis.